# shapiq: Shapley Interactions for Machine Learning

**Maximilian Muschalik**[*]
LMU Munich, MCML

**Hubert Baniecki**
University of Warsaw,
Warsaw University of Technology

**Fabian Fumagalli**
Bielefeld University

**Patrick Kolpaczki**
Paderborn University

**Barbara Hammer**
Bielefeld University

**Eyke Hüllermeier**
LMU Munich, MCML

## Abstract

Originally rooted in game theory, the Shapley Value (SV) has recently become an important tool in machine learning research. Perhaps most notably, it is used for feature attribution and data valuation in explainable artificial intelligence. Shapley Interactions (SIs) naturally extend the SV and address its limitations by assigning joint contributions to groups of entities, which enhance understanding of black box machine learning models. Due to the exponential complexity of computing SVs and SIs, various methods have been proposed that exploit structural assumptions or yield probabilistic estimates given limited resources. In this work, we introduce `shapiq`, an open-source Python package that unifies state-of-the-art algorithms to efficiently compute SVs and any-order SIs in an application-agnostic framework. Moreover, it includes a benchmarking suite containing 11 machine learning applications of SIs with pre-computed games and ground-truth values to systematically assess computational performance across domains. For practitioners, `shapiq` is able to explain and visualize any-order feature interactions in predictions of models, including vision transformers, language models, as well as XGBoost and LightGBM with TreeSHAP-IQ. With `shapiq`, we extend `shap` beyond feature attributions and consolidate the application of SVs and SIs in machine learning that facilitates future research. The source code and documentation are available at https://github.com/mmschlk/shapiq.

## 1 Introduction

Assigning *value* to entities collectively performing a task is essential in various real-world applications of machine learning (ML) [63, 74]. For instance, when reimbursing data providers based on the *value of data* [30, 80], or justifying a model's prediction based on *value of feature information* [13, 18, 19, 55, 77]. The *fair* distribution of value among a group of entities is a central aspect of cooperative game theory, where the Shapley Value (SV) [76] defines a *unique* allocation scheme based on intuitive axioms. The SV is applicable to any *game*, i.e. a function that specifies the worth of all possible groups of entities, called coalitions. In ML, application-specific games were introduced [5, 30, 74, 80, 84], which typically require a definition of the overall worth and a notion of entities' absence [19]. The SV fairly distributes the overall worth among individuals by evaluating the game for all coalitions. However, it does not give insights on *synergies or redundancies* between entities. For instance, while two features such as *latitude* and *longitude* convey separate information, only their joint consideration reveals the synergy of encoding an *exact location*. The value of such a group of entities is known as an *interaction* [33], or in this context *feature interaction* [29], and is crucial to understand predictions of complex ML models [20, 48, 49, 61, 66, 78, 82, 86], as illustrated in Figure 1.

---

[*]Corresponding author (maximilian.muschalik@ifi.lmu.de)

38th Conference on Neural Information Processing Systems (NeurIPS 2024) Track on Datasets and Benchmarks.

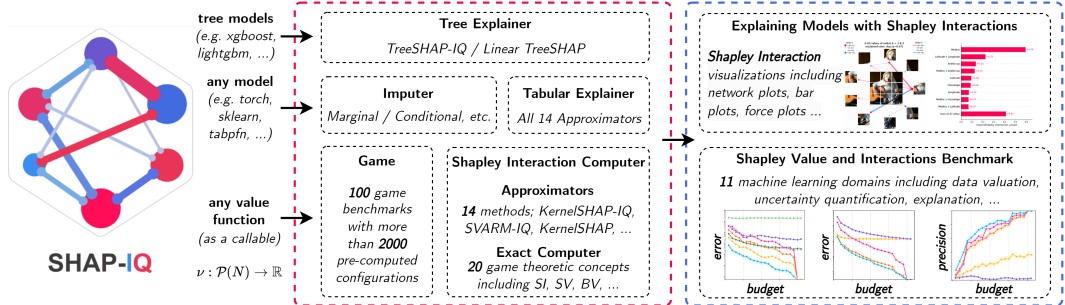

**Figure 1:** The `shapiq` Python package facilitates research on game theory for machine learning, including state-of-the-art approximation algorithms and pre-computed benchmarks. It provides a simple interface for explaining predictions of machine learning models beyond feature attributions **(1)**, enabling researchers to explore higher-order interactions or domain-independent game theory **(2)**.

Shapley Interactions (SIs) [10, 33, 78, 81] distribute the overall worth to all groups of entities up to a maximum *explanation order*. They satisfy axioms similar to the SV, to which they reduce for individuals, i.e. the lowest explanation order. In contrast, for the highest explanation order, which comprises an interaction for every coalition, the SIs yield the Möbius Interaction (MI), or Möbius transform [27, 35, 72]. The MIs are a fundamental concept in cooperative game theory that captures the *isolated joint contribution*, which allows to additively describe every coalition's worth by a sum of the contained MIs. With an increasing explanation order, the SIs comprise more components that finally yield the MIs as the most comprehensive explanation of the game at the cost of highest complexity [10, 81]. While the SV and SIs provide an appealing theoretical concept, computing them without structural assumptions on the game requires exponential complexity [22]. For tree-based models, it was shown that SVs [54, 87] and SIs [61, 88] can be efficiently computed by exploiting the architecture. Moreover, game-agnostic stochastic approximators estimate the SV [11, 17, 45, 55, 57, 64, 68] and SIs [28, 29, 47, 78, 81] with a limited budget of game evaluations.

Diverse applications of the SV have led to various techniques for its efficient computation [13]. Recently, extensions to any-order SIs addressed limitations of the SV and complemented interpretation of model predictions with higher-order feature interactions [10, 29, 78, 81, 82]. While stochastic approximators are applicable to any game, their evaluation is typically performed in an isolated application [29, 50], such as feature interactions. Moreover, implementing such algorithms requires a strong mathematical background and specific design choices. Existing Python packages, such as `shap` [55], provide a relatively small number of approximators, which are limited to the SV and feature attributions.

**Contribution.** In this work, we present `shapiq`, an open-source *Python* library for computing any-order SIs. `shapiq` consolidates research on SVs and SIs, making these tools accessible across various ML domains. Our core contributions are summarized in Figure 1 and include,

(1) a general `approximation` and computation interface for state-of-the-art SI algorithms and methods without focus on a specific application like explanations or data valuation,

(2) an explanation API for using SIs to explain ML models and visualizing interactions,

(3) a benchmarking suite to evaluate SI approximators across several real-world scenarios,

(4) and a cross-domain empirical evaluation of approximators guiding practitioners.

**Related software tools and benchmarks.** `shapiq` extends the popular `shap` [55] Python package beyond feature attributions aiming to fully embrace the application of SVs and SIs in ML. While `shap` implements a single index for 2-order feature interactions to explain the predictions of tree-based models, `shapiq` implements a dozen approximators for any-order SIs (Table 2) and offers a benchmarking suite for these algorithms across 10 different domains (Table 3). Related software such as `aix360` [4], `alibi` [42] and `dalex` [6] are general toolboxes offering the implementation and visualization of the most popular ML explanations for end-users. We specify in SIs to provide a comprehensive tool facilitating research in game theory for ML, including the exact computation of 20 interaction indices and game-theoretic concepts (Table 1). Notably, the `innvestigate` [3] and `captum` [44] Python packages offer feature attribution methods specific to (deep) neural networks. Most recently, `quantus` [36] implements evaluation metrics for these explanation methods.

We build upon recent advances in benchmarking explainable artificial intelligence (XAI) methods such as feature attributions [2, 51, 53, 63] and algorithms for data valuation [39]. `XAI-Bench` [53] focuses on synthetic tabular data. `OpenXAI` [2] provides 7 real-world tabular datasets with pre-trained neural network models, 7 feature attribution methods and 8 metrics to compare them. $\mathcal{M}^4$ [51] extends `OpenXAI` to benchmark feature attributions of deep neural networks for image and text modalities. In [63], the authors benchmark several algorithms for approximating SVs based on the conditional feature distribution. `OpenDataVal` [39] provides 9 real-world datasets, 11 data valuation methods and 4 metrics to compare them. `shapiq` puts more focus on benchmarking higher-order SI algorithms and provides an interface to state-of-the-art explanation methods that base on SIs, e.g. TreeSHAP-IQ [61]. While open data repositories such as `OpenML` [9] offer easy access to datasets for ML, we pre-compute and share ground-truth SIs for various games (i.e. dataset–model pairs) that saves considerable time and resources when benchmarking approximation algorithms.

## 2 Theoretical Background

In ML, various concepts are based on synergies of entities to optimize performance in a given task. For example, weak learners construct powerful model ensembles [73], collected data instances and features are used to train supervised ML models [16, 30], where feature values collectively predict outputs. To better understand such processes, XAI quantifies the contributions of these entities to the task, most prominently for feature values in predictions (local feature attribution [55, 77]), features in models (global feature importance [16, 17, 69]), and data instances in model training (data valuation [30]). Assigning such contributions is closely related to the field of cooperative game theory [74], which studies the notion of value for players that collectively obtain a payout. To adequately assess the impact of individual players, it is necessary to analyze the payout for different coalitions. More formally, a cooperative game $\nu : \mathcal{P}(N) \to \mathbb{R}$ with $\nu(\emptyset) = 0$ is defined by a *value function* on the power set of $N := \{1, \ldots, n\}$ entities, which describes such payouts for all possible coalitions of players. We later summarize such prominent examples in the context of ML in Table 3. Here, we summarize existing contribution concepts for individuals and groups of entities, outlined in Table 1.

The **SV** [76] and **Banzhaf Value (BV)** [8] are instances of *semivalues* [23]. Semivalues assign contributions to individual players and adhere to intuitive axioms: *Linearity* enforces linearly composed contributions for linearly composed games; *Dummy* requires that players without impact receive zero contribution; *Symmetry* enforces that entities contributing equally to the payout receive equal value. The SV [76] is the unique semivalue that additionally satisfies *efficiency*, i.e. the sum of all contributions yields the total payout $\nu(N)$. In contrast, the BV [8] is the unique semivalue that additionally satisfies *2-Efficiency*, i.e. the contributions of two players sum to the contribution of a

**Table 1:** Available concepts in the `ExactComputer` class in `shapiq` with **SIs** in **bold**.

| Setting | Interaction Index (II) [27] | Base Semivalue [23] | Generalized Value (GV) [58] |
|---|---|---|---|
| **Machine Learning** | **$k$-Shapley Values ($k$-SII) [10]** 
 **Shapley Taylor II (STII) [78]** 
 **Faithful Shapley II (FSII) [81]** 
 $k_{\text{ADD}}$-SHAP [68] 
 Faithful Banzhaf II (FBII) [81] | **Shapley (SV) [76]** 



 Banzhaf (BV) [8] | Joint SV [34] 



 – |
| **Game Theory** | **Möbius (MI) [27, 35, 72]** 
 Co-Möbius (Co-MI) [32] 
 Shapley II (SII) [33] 
 Chaining II (CHII) [60] 
 Banzhaf II (BII) [33] | – 

 **Shapley (SV) [76]** 
 Banzhaf (BV) [8] | Internal GV (IGV) [58] 
 External GV (EGV) [58] 
 Shapley GV (SGV) [59] 
 Chaining GV (CHGV) [58] 
 Banzhaf GV (BGV) [60] |

joint player in a reduced game, where both players are merged. The SV and BV are represented as a weighted average over *marginal contributions* $\Delta_i(T) := \nu(T \cup \{i\}) - \nu(T)$ for $i \in N$ as

$$\phi^{\text{SV}}(i) := \sum_{T \subseteq N \setminus \{i\}} \frac{1}{n \binom{n-1}{|T|}} \Delta_i(T) \qquad \text{and} \qquad \phi^{\text{BV}}(i) := \sum_{T \subseteq N \setminus \{i\}} \frac{1}{2^{n-1}} \Delta_i(T) \,.$$

In ML applications, the SV is typically preferred over the BV due to the efficiency axiom [74]. For instance, in local feature attribution, the SV is utilized to fairly distribute the model's prediction to individual features [55, 77]. However, it was shown that the SV is limited when explaining complex decision systems, and *feature interactions*, i.e. the joint contributions of features' groups, are required to understand such processes [20, 29, 48, 49, 61, 66, 78, 81, 82, 86].

The **Generalized Value (GV)** [58] and **Interaction Index (II)** [27] are two paradigms to extend the notion of value to groups of entities. The GVs are based on weighted averages over (joint) marginal contributions $\nu(T \cup S) - \nu(T)$ for $S \subseteq N$ given $T \subseteq N \setminus S$. In contrast, IIs are based on discrete derivatives that account for lower-order effects of subsets of $S$. For instance, for two players $i, j \in N$, the discrete derivative $\Delta_{\{i,j\}}(T)$ is defined as the joint marginal contribution $\nu(T \cup \{i, j\}) - \nu(T)$ minus the individual marginal contributions $\Delta_i(T)$ and $\Delta_j(T)$. More generally, the *discrete derivative* $\Delta_S(T)$ for $S \subseteq N$ in the presence of $T \subseteq N \setminus S$ is defined as

$$\Delta_S(T) := \sum_{L \subseteq S} (-1)^{|S|-|L|} \nu(T \cup L) \quad \text{with} \quad \Delta_S(T) = \underbrace{\nu(T \cup S) - \nu(T)}_{\text{joint marginal contribution}} - \sum_{\emptyset \neq L \subset S} \underbrace{\Delta_L(T)}_{\text{lower-order effects}} \,.$$

A positive value indicates synergy, whereas a negative value indicates redundancy of $S$ given $T$. Lastly, a zero value indicates (additive) independence, i.e. the joint marginal contribution is equal to the sum of all lower-order effects. GVs and IIs are uniquely represented [27, 58] by

$$\phi^{\text{GV}}(S) := \sum_{T \subseteq N \setminus S} p^{|S|}_{|T|}(n) \left( \nu(T \cup S) - \nu(T) \right) \qquad \text{and} \qquad \phi^{\text{II}}(S) := \sum_{T \subseteq N \setminus S} p^{|S|}_{|T|}(n) \Delta_S(T) \,.$$

The most prominent examples are the *Shapley GV (SGV)* [59] and the *Shapley II (SII)* [33] with $p^s_t(n) = \left( (n-s+1)\binom{n-s}{t} \right)^{-1}$, which naturally extend the SV (cf. Appendix A.1). While the SGV and SII are natural extensions to the SV, they are not suitable for interpretability, since they are defined on the powerset and comprise an exponential number of components. Moreover, neither GVs nor IIs satisfy the efficiency axiom for higher-orders, which is desirable for ML applications.

**Shapley Interactions (SIs) for Machine Learning** assign joint contribution $\Phi_k$ up to an *explanation order* $k$, i.e. for all coalitions $S \subseteq N$ with $|S| \leq k$, which satisfy generalized *efficiency* $\nu(N) = \sum_{S \subseteq N, |S| \leq k} \Phi_k(S)$. The *$k$-SVs ($k$-SIIs)* [10] are the unique SIs that coincide with SII for the highest order. The *Shapley Taylor II (STII)* [78] puts a stronger emphasis on the top-order interactions, and *Faithful SII (FSII)* [81] optimizes *Shapley-weighted faithfulness*

$$\mathcal{L}(\nu, \Phi_k) := \sum_{T \subseteq N} \mu(t) \left( \nu(T) - \sum_{S \subseteq T, |S| \leq k} \Phi_k(S) \right)^2 \quad \text{with} \quad \mu(t) := \begin{cases} \mu_\infty & \text{if } t \in \{0, n\} \\ \frac{1}{\binom{n-2}{t-1}} & \text{else} \end{cases} \,.$$

**Table 2:** Overview of methods in `shapiq` and applicable SIs. Explainers rely on approximators or model assumptions. (✓) indicates only top-order approximation.

| Class | Implementation | Source | SV | (k-)SII | STII | FSII |
|---|---|---|---|---|---|---|
| **Approximator** | SHAP-IQ | [29] | ✓ | ✓ | ✓ | (✓) |
| | SVARM-IQ | [47] | ✓ | ✓ | ✓ | (✓) |
| | Permutation Sampling (SII) | [81] | ✓ | ✓ | – | – |
| | Permutation Sampling (STII) | [78] | ✓ | – | ✓ | – |
| | KernelSHAP-IQ | [28] | ✓ | ✓ | – | – |
| | Inconsistent KernelSHAP-IQ | [28] | ✓ | ✓ | – | – |
| | FSII Regression | [81] | ✓ | – | – | ✓ |
| | KernelSHAP | [55] | ✓ | – | – | – |
| | $k_{ADD}$-SHAP | [68] | ✓ | – | – | – |
| | Unbiased KernelSHAP | [17] | ✓ | – | – | – |
| | SVARM | [45] | ✓ | – | – | – |
| | Permutation Sampling | [11] | ✓ | – | – | – |
| | Owen Sampling | [64] | ✓ | – | – | – |
| | Stratified Sampling | [57] | ✓ | – | – | – |
| **Explainer** | Agnostic (Baseline) | – | ✓ | ✓ | ✓ | ✓ |
| | Agnostic (Marginal) | – | ✓ | ✓ | ✓ | ✓ |
| | Agnostic (Conditional) | – | ✓ | ✓ | ✓ | ✓ |
| | TabPFN | [37, 75] | ✓ | ✓ | ✓ | ✓ |
| | TreeSHAP-IQ | [61] | ✓ | ✓ | ✓ | (✓) |
| | Linear TreeSHAP | [54, 87] | ✓ | – | – | – |
| **Computer** | Möbius Converter | – | ✓ | ✓ | ✓ | ✓ |
| | Exact Computer | – | ✓ | ✓ | ✓ | ✓ |

FSII is thus $\Phi_k^{\text{FSII}} := \arg\min_{\Phi_k} \mathcal{L}(\nu, \Phi_k)$, where $\mu_\infty \gg 1$ ensures efficiency. It was recently shown that pairwise SII and consequently $k$-SII with $k = 2$ optimize a faithfulness metric with slightly different weights [28]. For $k = 1$, all SIs reduce to the SV $\Phi_1 \equiv \phi^{\text{SV}}$, which minimizes faithfulness $\mathcal{L}(\nu, \Phi_1)$ [12] with the efficiency constraint, or equivalently $\mu_\infty \to \infty$ [28, 55]. Finally, for $k = n$, all SIs $\Phi_n$ are the *MIs* (cf. Appendix A.2), which are faithful to all game values, i.e. $\mathcal{L}(\nu, \Phi_n) = 0$. Notably, all SIs can be uniquely represented by the MIs [31]. In this context, SIs yield a complexity-accuracy trade-off, ranging from the least complex (SV) to the most comprehensive (MI) explanation. Other extensions include $k_{ADD}$-*SHAP* [68] of the SV and *Faithful BII (FBII)* [81] of the BV, which do not satisfy efficiency, as well as *Joint SVs* [34], a GV with efficiency. However, in the context of feature interactions and ML, SIs are preferred over GV-based (Joint SVs) or BV-based IIs (FBII), as they account for lower-order *interactions* and adhere to the SV and MI as edge cases.

# 3 Overview of the `shapiq` Python Package

The `shapiq` package accelerates research on SIs for ML, and provides an intutitve interface for explaining any-order feature interactions in predictions of ML models. Its code is open-source on GitHub at https://github.com/mmschlk/shapiq while the documentation with notebook examples and API reference is available at https://shapiq.readthedocs.io.

## 3.1 `shapiq` Facilitates Research on Shapley Interactions for Machine Learning

While SVs have been predominantly applied to explanation use cases by leveraging the `shap` [55] library, `shapiq` provides a more abstract perspective on cooperative games and allows researchers from various fields to interact with and extend the framework to incorporate SIs and SVs.

**Approximators.** We implement 7 algorithms for approximating SIs across 4 different interaction indices, and another 7 algorithms specifically for approximating SVs. Table 2 provides a comprehensive overview of this effort, where the `shapiq.Approximator` class is extended with each implementation. We unify common approximation methods by including a general `shapiq.CoalitionSampler`

```
X, model = ...
import shapiq
# create an explainer object
explainer = shapiq.Explainer(model=model, data=X, max_order=3)
# choose a sample point to be explained
x = X[0]
# approximate feature interactions given the specified budget
interaction_values = explainer.explain(x=x, budget=1024)
# retrieve 3-order feature interactions
interaction_values.get_n_order_values(3)
# visualize 1-order and 2-order feature interactions on a graph
interaction_values.plot_network(feature_names=...)
```

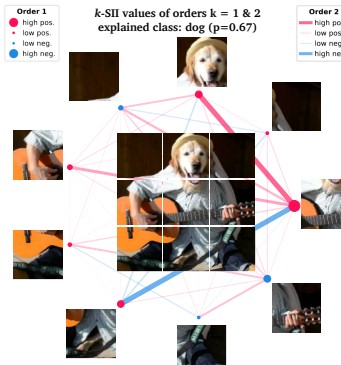

**Figure 2: Left:** Exemplary code for locally explaining a single model's prediction with `shapiq`. **Right:** Local feature interactions visualized on a network plot.

```
X, model = ...
import shapiq
# create an explainer object
explainer = shapiq.Explainer(model=model, data=X, max_order=3)
# approximate feature interactions for multiple sample points
interaction_values_list = explainer.explain_X(X, budget=1024)
# retrieve 3-order feature interactions for the first point
interaction_values_list[0].get_n_order_values(3)
# visualize the global feature interaction importance
shapiq.plot.bar_plot(interaction_values_list, feature_names=...)
```

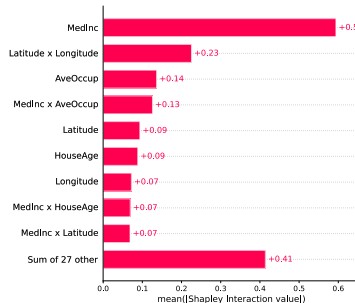

**Figure 3: Left:** Exemplary code for globally explaining multiple model's predictions with `shapiq`. **Right:** Global feature interaction importance visualized as a bar plot.

interface offering approximation performance increases through sampling procedures like the *border-* and *pairing-tricks* introduced in [17, 29].

**Exact computer.** A key functionality of `shapiq` lies in computing the SIs *exactly*, which is feasible for smaller games, but reaches its limit for growing player numbers. The `shapiq.ExactComputer` class provides an interface for computing 20 interaction indices and game-theoretic concepts, including the MI, BV, SGV, among others (see Table 1).

**Games.** Approximators and computers require the specification of a *cooperative game*. Games make up a central level of abstraction in `shapiq`, and specifying a game only requires the implementation of a domain-specific value function. Table 3 describes in detail 11 benchmark games implemented in `shapiq`. Beyond synthetic games, our benchmark spans the 5 most prominent domains where SIs can be applied for ML. The `shapiq.Game` class can be easily extended to include future benchmarks in the package. We pre-compute and share exact SIs for 2 042 benchmark game configurations in total (see Appendix B), facilitating future work on improving the approximators, which we elaborate on further in Section 4.

## 3.2 Explaining Machine Learning Predictions with `shapiq`

In addition to facilitating theoretical advancements, `shapiq` also provides practical tools for applying SIs in ML. These tools streamline the process of explaining feature interactions in model predictions, allowing researchers and practitioners to easily compute and visualize interaction effects across a range of models and data types.

**Explainer.** The `shapiq.Explainer` class is a simplified interface to explain any-order feature interactions in ML models. Figure 2 goes through exemplary code used to approximate SIs for a single prediction and visualize them on a graph plot. Currently three classes are further distinguished within the API, but we envision extending `shapiq.Explainer` to include more data modalities

**Table 3:** Overview of the available benchmark games and domains in `shapiq`. Each benchmark can be instantiated with different datasets, models, player sizes, or benchmark-specific configuration parameters. This results in $2\,042$ pre-computed individual configurations (see Tables 4 and 5).

| Domain | Benchmark (Game) | Source | Players | Coalition Worth |
|---|---|---|---|---|
| **XAI** | Local Explanation | [55, 77] | Features | Model Output |
| | Global Explanation | [18] | Features | Model Loss |
| | Tree Explanation | [54, 61] | Features | Model Output |
| **Uncertainty** | Uncertainty Explanation | [84] | Features | Prediction Entropy |
| **Model Selection** | Feature Selection | [16, 69] | Features | Performance |
| | Ensemble Selection | [73] | Weak Learners | Performance |
| | RF Ensemble Selection | [73] | Tree Models | Performance |
| **Valuation** | Data Valuation | [30] | Data Points | Performance |
| | Dataset Valuation | [80] | Data Subsets | Performance |
| **Unsupervised Learning** | Cluster Explanation | – | Features | Cluster Score |
| | Unsupervised Feature Importance | [5] | Features | Total Correlation |
| **Synthetic** | Sum of Unanimity Model | [28, 81] | Players | Sum of Unanimous Votes |

and model algorithms. `shapiq.TabularExplainer` allows for model-agnostic explanation based on feature marginalization with baseline, marginal, or conditional imputation (refer to Appendix C for details). `shapiq.TreeExplainer` implements TreeSHAP-IQ [61] for efficient explanations specific to decision tree-based models, e.g. random forest or gradient boosting decision trees, with native support for `scikit-learn` [67], `xgboost` [15], and `lightgbm` [40]. Figure 3 goes through exemplary code for explaining a set of predictions and visualizing their aggregation in a bar plot, which represents the global feature interaction importance. Lastly, `shapiq.TabPFNExplainer` is a special case of the `shapiq.TabularExplainer` handling explanations for the TabPFN [37] model architecture akin to [75] with a *remove-and-recontextualize* paradigm.

**Visualizing interactions.** `shapiq` supports the plotting and analysis of interaction values with different visualizations techniques. `shapiq.plot` offers custom visualizations including our custom network and SI graph plot, but also wraps established visualizations from `shap` [55] like the `force` and `bar` plots. For a detailed guide and summary of the visualizations, we kindly refer to Appendix D.

**Utility functions.** `shapiq` offers additional useful tools that are described in detail in the documentation. Interaction values are stored and processed using the `shapiq.InteractionValues` data class, which is rich in utility functions. Notably, useful set-based operators and generators exist for handling player sets $S \subseteq N$ or iterating over power sets $\mathcal{P}(N)$ of certain sizes with `shapiq.powerset`. Finally, `shapiq.datasets` loads datasets used for testing and examples.

## 4  Benchmarking Analysis

The `shapiq` library enables computation of various SIs for a broad class of application domains. To illustrate its versatility, we conduct benchmarks across a wide variety of traditional ML-based SV application scenarios. The ML benchmark demonstrates how higher-order SIs enable an accuracy–complexity trade-off for model interpretability (Section 4.1) and highlights that different approximation techniques in `shapiq` achieve the state-of-the-art performance depending on the application domains (Section 4.2). Tables 3, 4 and 5 present an overview of different application domains and associated benchmarks. Depending on the benchmark, it can be instantiated with different datasets, models, player numbers or benchmark-specific configuration parameters, e.g. uncertainty type: *epistemic* for Uncertainty Explanation or imputer: *conditional* for Local Explanation. In total, `shapiq` offers 100 unique benchmark games, i.e. applications times dataset–model pairs.

For all games that include $n \leq 16$ players, the value functions have been pre-computed by evaluating all coalitions and storing the games to file. Reading a pre-computed game from file, instead of performing up to $2^{16} = 65\,536$ value function calls with each new experiment run, saves valuable computational time and contributes to reproducibility as well as sustainability. This is particularly

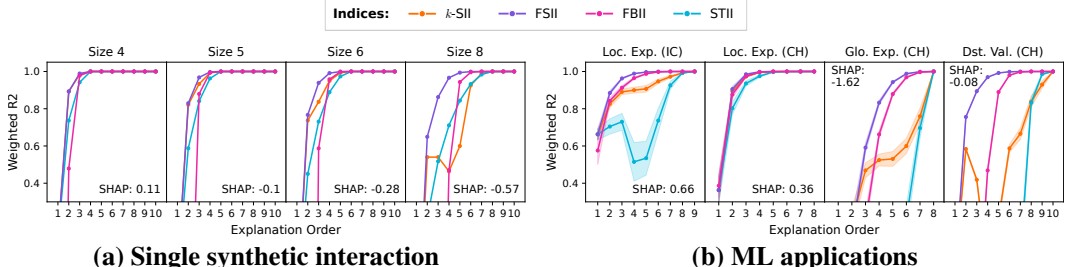

**Figure 4:** Shapley-weighted $R^2$ of interaction indices by explanation order for **(a)** single synthetic interactions and **(b)** ML applications. FSII is optimized for this metric, and increases faithfulness with each order. Interactions improve faithfulness over SHAP and yield an exact decomposition for the highest order. However, increasing interaction size negatively affects faithfulness.

beneficial for tasks that involve remove-and-refit strategies [19], such as Data Valuation or Feature Selection. For $n > 16$, where pre-computing a game and ground truth values becomes computationally prohibitive, we rely on analytical solutions to compute the ground truth like TreeSHAP-IQ [61] for tree-based ensembles or the MI representation of Sum of Unanimity Models [28, 29, 81]. For details regarding the experimental setting and reproducibility, we refer to Appendix E.

### 4.1 Faithfulness and Complexity of Shapley Interactions

In this experiment, we empirically assess how *faithfully* lower-order SIs representations capture higher-order effects for varying explanation orders $k$ and choice of interaction index. To this end, we rely on the Shapley-weighted faithfulness $\mathcal{L}(\nu, \Phi_k)$ introduced in Section 2. The complexity of SIs ranges from SVs (least complex) to MIs (most complex), where SVs minimize $\mathcal{L}(\nu, \Phi_1)$ for $k = 1$ [12, 55], while MIs perfectly capture all game values with $\mathcal{L}(\nu, \Phi_n) = 0$ for $k = n$ [10, 81]. To quantify the faithfulness of SIs across different domains, we calculate interaction values for a given index $\Phi_k$ and explanation order $k$. We then approximate the game values for each subset $T \subseteq N$ of players as $\hat{\nu}_k(T) := \sum_{S \subseteq T : |S| \leq k} \Phi_k(S)$ and compare them to the ground truths using $\mathcal{L}(\nu, \Phi_k)$ through a Shapley-weighted $R^2 (\nu(T), \hat{\nu}_k(T))$ for all $T \subseteq N$. For context, a game with no interactions (a 1-additive game) will be perfectly reproduced by a 1-additive explanation, yielding $\mathcal{L}(\nu, \Phi_1) = 0$ and $R^2 = 1$. Conversely, games with substantial higher-order interactions will result in larger errors for lower-order explanations, with $\mathcal{L}(\nu, \Phi_k) \gg 0$ and $R^2 < 1$.

Figure 4 shows the Shapley-weighted $R^2$ value for $k = 1, \ldots, n$ for a synthetic game with a single interaction of varying size (a) and real-world ML applications (b). Here, we used $\mu_\infty = 1$ instead of $\mu_\infty \gg 1$, which affects FBII that violates efficiency. The results show that in general SIs become more faithful with higher explanation order. Notably, the difference between pairwise SIs and SVs (SHAP textbox) is remarkable, where pairwise interactions ($k = 2$) already yield a strong improvement in faithfulness. If higher-order interactions dominate, then SIs require a larger explanation order to maintain faithfulness. While FSII and FBII are optimized for faithfulness, STII and $k$-SII do not necessarily yield a strict improvement in this metric. In fact, it was shown that SII and $k$-SII optimize a slightly different faithfulness metric, which changes for every order [28]. Yet, we observe a consistent strong improvement of pairwise $k$-SII over the SV (SHAP). While FSII and FBII optimize faithfulness, $k$-SII and STII adhere to strict structural assumptions, where STII projects all higher-order interactions to the top-order SIs, and $k$-SII is consistent with SII. Practitioners may choose SIs tailored to their specific application, where $k$-SII is a good default choice for `shapiq`.

### 4.2 Comparison of Approximation Methods

Various approximation methods for computing SIs are included in `shapiq` for a variety of SIs (cf. Table 2). The possibility of attributing (domain-specific) state-of-the-art performance to a single algorithm has been investigated empirically by multiple works [28, 29, 45–47, 57, 64, 68, 81, 83, 89]. We use the collection of 100 unique benchmark games in `shapiq` to evaluate the performance of different SV and SI approximation methods on a broad spectrum of ML applications. For each domain and configuration (see Table 4 and 5 in Appendix B), we compute ground truth SVs, 2-SIIs,

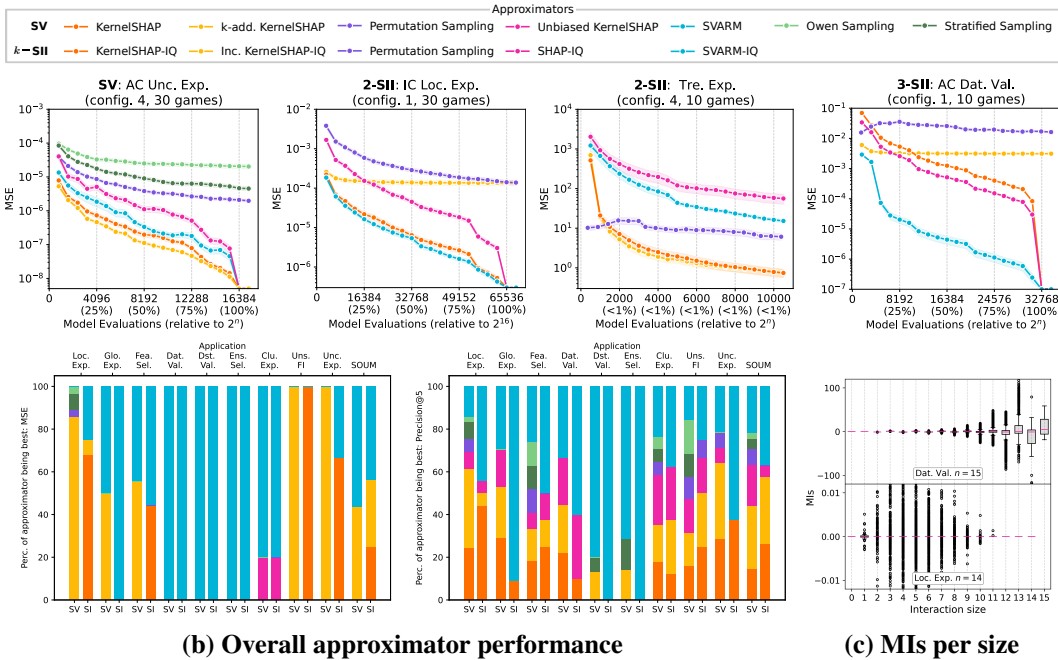

**(a) Approximation quality in different application domains**

**(b) Overall approximator performance**      **(c) MIs per size**

**Figure 5:** Overview of the benchmark results containing **(a)** budget-dependent MSE approximation curves on different benchmark settings, **(b)** a summary of the best performing approximators per setting over all 100 benchmark games measured by MSE (left) and Precision@5 (right), and **(c)** exemplary MIs for ten games of Data Valuation (top) and Local Explanation (bottom).

and 3-SIIs and compare them with estimates provided by all approximators from Table 2. The approximators are run with a wide range of budget values and assessed by their achieved mean squared error (MSE) or precision at five (Precision@5). Figure 5 summarizes the approximation results.

Most notably, the ranking of approximators varies strongly between the different applications domains, which is depicted in Figure 5 (a) and (b). This observation holds for both SVs and SIs. In general, two types of approximation methods dominate the application landscape in terms of MSE and Precision@5. First, *kernel-based* approaches including KernelSHAP, $k_{\mathrm{ADD}}$-SHAP, KernelSHAP-IQ and Inconsistent KernelSHAP-IQ perform best for Local Explanation, Uncertainty Explanation, and Unsupervised Feature Importance. Second, the two *stratification-based* estimators SVARM and SVARM-IQ achieve state-of-the-art performance for Data Valuation, Dataset Valuation, or Ensemble Selection. Traditional *mean-estimation* methods including Permutation Sampling (SV and SII), Unbiased KernelSHAP, SHAPIQ, and Owen Sampling achieve moderate estimation qualities in comparison. Our findings give rise to the conclusion that *stratification-based* estimators perform superior in settings where the size of a coalition naturally impacts its worth (e.g. training size for Dataset Valuation), which is plausible as these methods group coalitions by size and thus leverage this dependency. Meanwhile, *kernel-based* estimators achieve state-of-the-art in settings where the dependency between size and worth of a coalition is less pronounced (e.g. sudden jumps of model predictions in Local Explanation).

Interestingly, the settings where *stratified-sampling* outperforms *kernel-based* variants exhibit different internal structures in the games' MIs. Generally, MIs disentangle a game into all of its additive components (cf. Section 2) and can be computed exactly with `shapiq`'s pre-computed games. The accuracy of *kernel-based* estimators drops when higher-order interactions dominate the games' structure instead of lower-order interactions. This is depicted by Figure 5 (c) where the MIs for Local Explanation are of lower order than the Data Valuation games.

# 5 Conclusion

As SIs are increasingly employed to analyze ML models, it becomes pivotal to ensure that these are accurately and efficiently approximated. To this end, we contributed `shapiq`, an open-source toolbox that implements state-of-the-art algorithms, defines a dozen of benchmark games, and provides ready-to-use explanations of any-order feature interactions. `shapiq` contains a comprehensive documentation and is designed to be extendable by contributors.

**Limitations and future work.** We identify three main limitations of `shapiq` that provide natural opportunities for future work. First, the TreeSHAP-IQ algorithm is currently implemented in Python, but by-design requires no access to model inference, which allows for a more efficient implementation in C++ alike TreeSHAP [54, 87]. Second, SIs can be misinterpreted based on choosing the wrong index for the application scenario, which we comment on across Sections 2 and 4.1. The selection of a particular SI index, enabled by `shapiq`, offers great opportunities for application-specific research. We also acknowledge that visualization of higher-order feature interactions is itself challenging and a potential research direction in human-computer interaction. Certainly, a human-centric evaluation of explanations may be required for their broader adoption in practical applications [71].

**Broader impact.** A potential negative societal implication of visualizing higher-order feature interactions may be an *information overload* [7, 70] that leads to users misinterpreting model explanations. Nevertheless, we hope our contribution sparks the advancement of game-theoretical indices motivated by various applications in ML. Specifically in the context of explainability, `shapiq` may impact the way users interact with ML models when having access to previously inaccessible information, e.g. higher-order feature interactions.

## Acknowledgments and Disclosure of Funding

We gratefully thank the anonymous reviewers for their valuable feedback for improving this work. We also gratefully thank Santo Thies for supporting the implementation. Fabian Fumagalli and Maximilian Muschalik gratefully acknowledge funding by the Deutsche Forschungsgemeinschaft (DFG, German Research Foundation): TRR 318/1 2021 – 438445824. Hubert Baniecki was supported from the state budget within the Polish Ministry of Education and Science program "Pearls of Science" project number PN/01/0087/2022. Patrick Kolpaczki was supported by the research training group Dataninja (Trustworthy AI for Seamless Problem Solving: Next Generation Intelligence Joins Robust Data Analysis) funded by the German federal state of North Rhine-Westphalia.

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

## Organization of the Supplementary Material

The supplementary material is organized as follows:

# A  Extended Theoretical Background

In this section, we introduce further theoretical background. Specifically, we discuss in more detail the class of GVs and IIs in Appendix A.1, and the MIs in Appendix A.2.

## A.1  Probabilistic and Cardinal-Probabilistic GVs and IIs

Probabilistic GVs [58] extend semivalues with a focus on *monotonicity*, i.e. games that satisfy $\nu(S) \leq \nu(T)$, if $S \subseteq T \subseteq N$. GVs satisfy the *positivity axiom*, which requires non-negative joint contributions, i.e. $\phi^{\mathrm{GV}}(S) \geq 0$, for all $S \subseteq N$ in monotone games [58]. It was shown that GVs are uniquely represented as weighted averages over (joint) marginal contributions $\nu(T \cup S) - \nu(T)$. On the other hand, cardinal-probabilistic IIs [27] are centered around synergy, independence and redundancy between entities. IIs are based on discrete derivatives, which extend (joint) marginal contributions by accounting for lower-order effects. IIs focus on $k$-*monotonicity*, i.e. games that have non-negative discrete derivatives $\Delta_S(T) \geq 0$ for $S \subseteq U \subseteq N$ with $2 \leq |S| \leq k$. IIs satisfy the $k$-*monotonicity axiom*, i.e. non-negative interactions $\phi^{\mathrm{II}}(S) \geq 0$ for $k$-monotone games. Both, GVs and IIs are uniquely represented [27, 58] as

$$\phi^{\mathrm{GV}}(S) := \sum_{T \subseteq N \setminus S} p_{|T|}^{|S|}(n) \cdot (\nu(T \cup S) - \nu(T)) \quad \text{and} \quad \phi^{\mathrm{II}}(S) := \sum_{T \subseteq N \setminus S} p_{|T|}^{|S|}(n) \cdot \Delta_S(T),$$

where $p_t^s(n)$ are index-specific weights based on the sizes of $S, T$ and $N$. The *SGV* [59] and the *SII* [33] with

$$\textbf{Shapley: } p_t^s(n) = \frac{1}{n - s + 1} \binom{n - s}{t}^{-1}$$

naturally extend the SV. An alternative extension for the SV is the *Chaining GV (CHGV)* [58] and *Chaining II (CHII)* [60] with

$$\textbf{Chaining: } p_t^s(n) = \frac{s}{s + t} \binom{n}{s + t}^{-1}.$$

The main difference of the SGV/SII and the CHGV/CHII is the quantification of so-called *partnerships* [27, 58], i.e. coalitions that only influence the value of the game if all members of the partnership are present. The CHGV and CHII adhere to the *partnership-allocation axiom* [27, 58], which states that the contribution of an individual member of the partnership and the interaction of the whole partnership are proportional. In contrast, the SGV and SII satisfy the *reduced partnership consistency axiom* [27, 58], which states that the interaction of the whole partnership is equal to the contribution of the partnership in a game, where the partnership is a single player.

On the other hand, the *Banzhaf GV (BGV)* [59] and *Banzhaf II (BII)* [33] extend the BV with

$$\textbf{Banzhaf: } p_t^s(n) := \frac{1}{2^{n-s}}.$$

## A.2  Möbius Interactions (MIs)

The MIs $\Phi_n$ are a prominent concept in discrete mathematics, which appears in many different forms. In discrete mathematics, it is also known as the Möbius transform [72]. In cooperative game theory, the concept is known as Harsanyi dividend [35] or internal II [27]. The MI for $S \subseteq N$ is defined as

$$\Phi_n(S) := \sum_{T \subseteq S} (-1)^{|S| - |T|} \nu(T).$$

In this context, the MIs are the unique measure that satisfy the *recovery property*

$$\nu(T) = \sum_{S \subseteq T} \Phi_n(S) \text{ for every } T \subseteq N.$$

The MIs are a basis of the vector space of cooperative games, and thus every game can be uniquely represented in terms of its MIs. The *Co-Möbius transform (Co-MI)* [32] is another fundamental concept linked to the MIs of the conjugate game, i.e. $\bar{\nu}(T) := \nu(N \setminus T)$ [31].

# B Detailed Overview of all Benchmark Games and Configurations

## B.1 Benchmark Overview

We list in Table 4 and 5 all configurations available within our benchmark. Depending on the application task, a configuration represents a combination of multiple parameters that specify the generated cooperative games. For ML games, such a combination includes at least the used dataset and a number of features or datapoints. If a prediction model or imputer for feature values is employed, as for example for Local XAI games, these are also specified.

**Table 4:** Overview of all Benchmark Configurations: Each configuration is assigned a distinctive identifier (ID), has a name (Benchmark) indicating dataset and application if available, is pre-computed (P.) if the player number $n$ does not exceed 16, consists of $|\mathcal{P}(N)|$ many coalitions to be evaluated, is iterated over multiple game instances ($g$), and has a set of parameters (Game Configuration).

| ID | Benchmark | Data | P. | $n$ | $|\mathcal{P}(N)|$ | $g$ | Game Configuration |
|---|---|---|---|---|---|---|---|
| 1 | Data Valuation | AD | ✓ | 15 | 32768 | 10 | model_name=decision_tree, n_data_points=15 |
| 2 | Data Valuation | AD | ✓ | 15 | 32768 | 10 | model_name=random_forest, n_data_points=15 |
| 3 | Data Valuation | AD | ✓ | 10 | 1024 | 30 | model_name=decision_tree, player_sizes=increasing, n_players=10 |
| 4 | Data Valuation | AD | ✓ | 10 | 1024 | 30 | model_name=random_forest, player_sizes=increasing, n_players=10 |
| 5 | Data Valuation | AD | ✓ | 10 | 1024 | 30 | model_name=gradient_boosting, player_sizes=increasing, n_players=10 |
| 6 | Data Valuation | AD | ✓ | 14 | 16384 | 5 | model_name=decision_tree, player_sizes=increasing, n_players=14 |
| 7 | Ensemble Selection | AD | ✓ | 10 | 1024 | 30 | loss_function=accuracy_score, n_members=10 |
| 8 | Feature Selection | AD | ✓ | 14 | 16384 | 30 | model_name=decision_tree |
| 9 | Feature Selection | AD | ✓ | 14 | 16384 | 30 | model_name=random_forest |
| 10 | Feature Selection | AD | ✓ | 14 | 16384 | 30 | model_name=gradient_boosting |
| 11 | Global Explanation | AD | ✓ | 14 | 16384 | 30 | model_name=decision_tree, loss_function=accuracy_score |
| 12 | Global Explanation | AD | ✓ | 14 | 16384 | 30 | model_name=random_forest, loss_function=accuracy_score |
| 13 | Global Explanation | AD | ✓ | 14 | 16384 | 30 | model_name=gradient_boosting, loss_function=accuracy_score |
| 14 | Local Explanation | AD | ✓ | 14 | 16384 | 30 | model_name=decision_tree, imputer=marginal |
| 15 | Local Explanation | AD | ✓ | 14 | 16384 | 30 | model_name=random_forest, imputer=marginal |
| 16 | Local Explanation | AD | ✓ | 14 | 16384 | 30 | model_name=gradient_boosting, imputer=marginal |
| 17 | Local Explanation | AD | ✓ | 14 | 16384 | 30 | model_name=decision_tree, imputer=conditional |
| 18 | Local Explanation | AD | ✓ | 14 | 16384 | 30 | model_name=random_forest, imputer=conditional |
| 19 | Local Explanation | AD | ✓ | 14 | 16384 | 30 | model_name=gradient_boosting, imputer=conditional |
| 20 | RF Ensemble Selection | AD | ✓ | 10 | 1024 | 30 | loss_function=accuracy_score, n_members=10 |
| 21 | Uncertainty Explanation | AD | ✓ | 14 | 16384 | 30 | uncertainty_to_explain=total, imputer=marginal |
| 22 | Uncertainty Explanation | AD | ✓ | 14 | 16384 | 30 | uncertainty_to_explain=total, imputer=conditional |
| 23 | Uncertainty Explanation | AD | ✓ | 14 | 16384 | 30 | uncertainty_to_explain=aleatoric, imputer=marginal |
| 24 | Uncertainty Explanation | AD | ✓ | 14 | 16384 | 30 | uncertainty_to_explain=aleatoric, imputer=conditional |
| 25 | Uncertainty Explanation | AD | ✓ | 14 | 16384 | 30 | uncertainty_to_explain=epistemic, imputer=marginal |
| 26 | Uncertainty Explanation | AD | ✓ | 14 | 16384 | 30 | uncertainty_to_explain=epistemic, imputer=conditional |
| 27 | Unsupervised FI. | AD | ✓ | 14 | 16384 | 1 | - |
| 28 | Cluster Explanation | BS | ✓ | 12 | 4096 | 1 | cluster_method=kmeans, score_method=silhouette_score |
| 29 | Cluster Explanation | BS | ✓ | 12 | 4096 | 1 | cluster_method=agglomerative, score_method=calinski_harabasz_score |
| 30 | Data Valuation | BS | ✓ | 15 | 32768 | 10 | model_name=decision_tree, n_data_points=15 |
| 31 | Data Valuation | BS | ✓ | 15 | 32768 | 10 | model_name=random_forest, n_data_points=15 |
| 32 | Data Valuation | BS | ✓ | 10 | 1024 | 30 | model_name=decision_tree, player_sizes=increasing, n_players=10 |
| 33 | Data Valuation | BS | ✓ | 10 | 1024 | 30 | model_name=random_forest, player_sizes=increasing, n_players=10 |
| 34 | Data Valuation | BS | ✓ | 10 | 1024 | 30 | model_name=gradient_boosting, player_sizes=increasing, n_players=10 |
| 35 | Data Valuation | BS | ✓ | 14 | 16384 | 5 | model_name=decision_tree, player_sizes=increasing, n_players=14 |
| 36 | Ensemble Selection | BS | ✓ | 10 | 1024 | 30 | loss_function=r2_score, n_members=10 |
| 37 | Feature Selection | BS | ✓ | 12 | 4096 | 30 | model_name=decision_tree |
| 38 | Feature Selection | BS | ✓ | 12 | 4096 | 30 | model_name=random_forest |
| 39 | Feature Selection | BS | ✓ | 12 | 4096 | 30 | model_name=gradient_boosting |
| 40 | Global Explanation | BS | ✓ | 12 | 4096 | 30 | model_name=decision_tree, loss_function=r2_score |
| 41 | Global Explanation | BS | ✓ | 12 | 4096 | 30 | model_name=random_forest, loss_function=r2_score |
| 42 | Global Explanation | BS | ✓ | 12 | 4096 | 30 | model_name=gradient_boosting, loss_function=r2_score |
| 43 | Local Explanation | BS | ✓ | 12 | 4096 | 30 | model_name=decision_tree, imputer=marginal |
| 44 | Local Explanation | BS | ✓ | 12 | 4096 | 30 | model_name=random_forest, imputer=marginal |
| 45 | Local Explanation | BS | ✓ | 12 | 4096 | 30 | model_name=gradient_boosting, imputer=marginal |
| 46 | Local Explanation | BS | ✓ | 12 | 4096 | 30 | model_name=decision_tree, imputer=conditional |
| 47 | Local Explanation | BS | ✓ | 12 | 4096 | 30 | model_name=random_forest, imputer=conditional |
| 48 | Local Explanation | BS | ✓ | 12 | 4096 | 30 | model_name=gradient_boosting, imputer=conditional |
| 49 | RF Ensemble Selection | BS | ✓ | 10 | 1024 | 30 | loss_function=r2_score, n_members=10 |
| 50 | Unsupervised FI. | BS | ✓ | 12 | 4096 | 1 | – |

**Table 5:** Overview of all Benchmark Configurations: Each configuration is assigned a distinctive identifier (ID), has a name (Benchmark) indicating dataset and application if available, is pre-computed (P.) if the player number $n$ does not exceed 16, consists of $|\mathcal{P}(N)|$ many coalitions to be evaluated, is iterated over multiple game instances ($g$), and has a set of parameters (Game Configuration).

| ID | Benchmark | Data | P. | $n$ | $|\mathcal{P}(N)|$ | $g$ | Game Configuration |
|---|---|---|---|---|---|---|---|
| 51 | Cluster Explanation | CH | ✓ | 8 | 256 | 1 | cluster_method=kmeans, score_method=silhouette_score |
| 52 | Cluster Explanation | CH | ✓ | 8 | 256 | 1 | cluster_method=agglomerative, score_method=calinski_harabasz_score |
| 53 | Data Valuation | CH | ✓ | 15 | 32768 | 10 | model_name=decision_tree, n_data_points=15 |
| 54 | Data Valuation | CH | ✓ | 15 | 32768 | 10 | model_name=random_forest, n_data_points=15 |
| 55 | Data Valuation | CH | ✓ | 10 | 1024 | 30 | model_name=decision_tree, player_sizes=increasing, n_players=10 |
| 56 | Data Valuation | CH | ✓ | 10 | 1024 | 30 | model_name=random_forest, player_sizes=increasing, n_players=10 |
| 57 | Data Valuation | CH | ✓ | 10 | 1024 | 30 | model_name=gradient_boosting, player_sizes=increasing, n_players=10 |
| 58 | Data Valuation | CH | ✓ | 14 | 16384 | 5 | model_name=decision_tree, player_sizes=increasing, n_players=14 |
| 59 | Ensemble Selection | CH | ✓ | 10 | 1024 | 30 | loss_function=r2_score, n_members=10 |
| 60 | Feature Selection | CH | ✓ | 8 | 256 | 30 | model_name=decision_tree |
| 61 | Feature Selection | CH | ✓ | 8 | 256 | 30 | model_name=random_forest |
| 62 | Feature Selection | CH | ✓ | 8 | 256 | 30 | model_name=gradient_boosting |
| 63 | Global Explanation | CH | ✓ | 8 | 256 | 30 | model_name=decision_tree, loss_function=r2_score |
| 64 | Global Explanation | CH | ✓ | 8 | 256 | 30 | model_name=random_forest, loss_function=r2_score |
| 65 | Global Explanation | CH | ✓ | 8 | 256 | 30 | model_name=gradient_boosting, loss_function=r2_score |
| 66 | Global Explanation | CH | ✓ | 8 | 256 | 30 | model_name=neural_network, loss_function=r2_score |
| 67 | Local Explanation | CH | ✓ | 8 | 256 | 30 | model_name=decision_tree, imputer=marginal |
| 68 | Local Explanation | CH | ✓ | 8 | 256 | 30 | model_name=random_forest, imputer=marginal |
| 69 | Local Explanation | CH | ✓ | 8 | 256 | 30 | model_name=gradient_boosting, imputer=marginal |
| 70 | Local Explanation | CH | ✓ | 8 | 256 | 30 | model_name=neural_network, imputer=marginal |
| 71 | Local Explanation | CH | ✓ | 8 | 256 | 30 | model_name=decision_tree, imputer=conditional |
| 72 | Local Explanation | CH | ✓ | 8 | 256 | 30 | model_name=random_forest, imputer=conditional |
| 73 | Local Explanation | CH | ✓ | 8 | 256 | 30 | model_name=gradient_boosting, imputer=conditional |
| 74 | Local Explanation | CH | ✓ | 8 | 256 | 30 | model_name=neural_network, imputer=conditional |
| 75 | RF Ensemble Selection | CH | ✓ | 10 | 1024 | 30 | loss_function=r2_score, n_members=10 |
| 76 | Unsupervised FI. | CH | ✓ | 8 | 256 | 1 | – |
| 77 | Local Explanation | IC | ✓ | 14 | 16384 | 30 | model_name=resnet_18, n_superpixel_resnet=14 |
| 78 | Local Explanation | IC | ✓ | 9 | 512 | 30 | model_name=vit_9_patches |
| 79 | Local Explanation | IC | ✓ | 16 | 65536 | 30 | model_name=vit_16_patches |
| 80 | Sum of Unanimity Model | Syn | ✓ | 15 | 32768 | 10 | n=15, n_basis_games=30, min_interaction_size=1, max_interaction_size=5 |
| 81 | Sum of Unanimity Model | Syn | ✓ | 15 | 32768 | 10 | n=15, n_basis_games=30, min_interaction_size=1, max_interaction_size=15 |
| 82 | Sum of Unanimity Model | Syn | ✓ | 15 | 32768 | 10 | n=15, n_basis_games=150, min_interaction_size=1, max_interaction_size=5 |
| 83 | Sum of Unanimity Model | Syn | ✓ | 15 | 32768 | 10 | n=15, n_basis_games=150, min_interaction_size=1, max_interaction_size=15 |
| 84 | Sum of Unanimity Model | Syn | X | 30 | $> 2^{16}$ | 10 | n=30, n_basis_games=30, min_interaction_size=1, max_interaction_size=5 |
| 85 | Sum of Unanimity Model | Syn | X | 30 | $> 2^{16}$ | 10 | n=30, n_basis_games=30, min_interaction_size=1, max_interaction_size=15 |
| 86 | Sum of Unanimity Model | Syn | X | 30 | $> 2^{16}$ | 10 | n=30, n_basis_games=30, min_interaction_size=1, max_interaction_size=25 |
| 87 | Sum of Unanimity Model | Syn | X | 30 | $> 2^{16}$ | 10 | n=30, n_basis_games=150, min_interaction_size=1, max_interaction_size=5 |
| 88 | Sum of Unanimity Model | Syn | X | 30 | $> 2^{16}$ | 10 | n=30, n_basis_games=150, min_interaction_size=1, max_interaction_size=15 |
| 89 | Sum of Unanimity Model | Syn | X | 30 | $> 2^{16}$ | 10 | n=30, n_basis_games=150, min_interaction_size=1, max_interaction_size=25 |
| 90 | Sum of Unanimity Model | Syn | X | 50 | $> 2^{16}$ | 10 | n=50, n_basis_games=30, min_interaction_size=1, max_interaction_size=5 |
| 91 | Sum of Unanimity Model | Syn | X | 50 | $> 2^{16}$ | 10 | n=50, n_basis_games=30, min_interaction_size=1, max_interaction_size=15 |
| 92 | Sum of Unanimity Model | Syn | X | 50 | $> 2^{16}$ | 10 | n=50, n_basis_games=30, min_interaction_size=1, max_interaction_size=25 |
| 93 | Sum of Unanimity Model | Syn | X | 50 | $> 2^{16}$ | 10 | n=50, n_basis_games=150, min_interaction_size=1, max_interaction_size=5 |
| 94 | Sum of Unanimity Model | Syn | X | 50 | $> 2^{16}$ | 10 | n=50, n_basis_games=150, min_interaction_size=1, max_interaction_size=15 |
| 95 | Sum of Unanimity Model | Syn | X | 50 | $> 2^{16}$ | 10 | n=50, n_basis_games=150, min_interaction_size=1, max_interaction_size=25 |
| 96 | Local Explanation | MR | ✓ | 14 | 16384 | 30 | mask_strategy=mask |
| 97 | Tree Explanation | Syn | X | 30 | $> 2^{16}$ | 10 | model_name=decision_tree, classification=True, n_features=30 |
| 98 | Tree Explanation | Syn | X | 30 | $> 2^{16}$ | 10 | model_name=random_forest, classification=True, n_features=30 |
| 99 | Tree Explanation | Syn | X | 30 | $> 2^{16}$ | 10 | model_name=decision_tree, classification=False, n_features=30 |
| 100 | Tree Explanation | Syn | X | 30 | $> 2^{16}$ | 10 | model_name=random_forest, classification=False, n_features=30 |

## B.2 Datasets and Models Used

Our benchmark games are based on five datasets. All of these datasets are publicly available. The following contains a small description of all datasets:

**AC:** The *AdultCensus* [43, CC BY 4.0 license] dataset is a tabular classification dataset containing $n = 14$ features. The dataset was obtained from `openml` [25] (id: *1590*) and is available at `https://github.com/mmschlk/shapiq/blob/v1/data/adult_census.csv` for reproducibility.

**BS:** The *BikeSharing* [24, CC BY 4.0 license] dataset is a tabular regression dataset containing $n = 12$ features. The dataset was obtained from `openml` [25] (id: *42712*) and is available at `https://github.com/mmschlk/shapiq/blob/v1/data/bike.csv` for reproducibility.

**CH:** The *CaliforniaHousing* [41, CC0 public domain] dataset is a tabular regression dataset containing $n = 8$ features. The target of this dataset is to predict property prices. The dataset was obtained from `scikit-learn` [67] and is available at `https://github.com/mmschlk/shapiq/blob/v1/data/california_housing.csv` for reproducibility.

**MR:** The *MovieReview* is also known as the IMBD dataset [56, custom research license] contains movie review excerpts. We simplify the dataset to only contain sentence parts with $n \leq 14$ words. The simplified dataset can be found at `https://github.com/mmschlk/shapiq/blob/v1/benchmark/data/simplified_imdb.csv` for reproducibility.

**IC:** The *ImageClassification* data contains test images from *Imagenet* [21, custom research license]. The example images can be found at `https://github.com/mmschlk/shapiq/tree/v1/shapiq/games/benchmark/imagenet_examples` for reproducibility.

All models used for the benchmark games are defined in the code repository. We use decision tree, random forest, k-nearest neighbour, linear/logistic regression models from `scikit-learn` [67]. Moreover, we use gradient-boosted tree classifiers and regressors from `xgboost` [15]. We train small neural networks with `PyTorch` [65] and use `PyTorch`'s `ResNet18` architecture. The movie review language model and the vision transformer are derived from the `transformers` API [85].

## B.3 Benchmarking Approximators of SIs with `shapiq`

Listing 1 shows an API for benchmarking 4 approximation algorithms on a Dataset Valuation game based on the *AdultCensus* dataset and a gradient boosting decision tree model.[2]

```python
import shapiq
from shapiq.benchmark import (
    load_games_from_configuration,
    print_benchmark_configurations,
    plot_approximation_quality,
    run_benchmark
)

# print all available games and benchmark configurations
print_benchmark_configurations()
>> Game: AdultCensusDatasetValuation
>> Player ID: 0
>> Number of Players: 10
>> Number of configurations: 3
>> Is the Benchmark Pre-computed: True
>> Iteration Parameter: random_state
>> Configurations:
>> Configuration 1: {'model_name': 'decision_tree', 'player_sizes': 'increasing', 'n_players': 10}
>> Configuration 2: {'model_name': 'random_forest', 'player_sizes': 'increasing', 'n_players': 10}
>> Configuration 3: {'model_name': 'gradient_boosting', 'player_sizes': 'increasing', 'n_players': 10}
>> ...

# load the game files from disk / or download
games = load_games_from_configuration(game_class="AdultCensusDataValuation", n_player_id=0, config_id=2)
games = list(games)  # convert to list (the generator is consumed)
n_players = games[0].n_players

# define the approximators to benchmark
sv_approximators = [
    shapiq.PermutationSamplingSII(n=n_players, index="k-SII", random_state=0),
    shapiq.SHAPIQ(n=n_players, random_state=0),
    shapiq.SVARMIQ(n=n_players, random_state=0),
    shapiq.KernelSHAPIQ(n=n_players, random_state=0)
]

# run the benchmark with the chosen parameters
results = run_benchmark(
    index="k-SII",
    order=2,
    games=games,
    approximators=sv_approximators,
    save_path="benchmark_results.json",
    budget_steps=[500, 1000, 2000, 4000],
    n_jobs=8
)

# plot the results
plot_approximation_quality(results)
```

**Listing 1:** Exemplary code for benchmarking approximators with `shapiq`.

---

[2]For details, refer to the notebook examples at `https://shapiq.readthedocs.io` referring to the benchmark capabilities.

## C  Marginal and Conditional Imputers

When computing SVs and SIs, especially for structured tabular data that has a natural interpretation of feature distribution, there is a choice for marginalizing feature influence over either a marginal or a conditional distribution [1, 14, 18, 52, 54, 55, 62, 63, 79].

For a concrete example [52], consider a supervised learning task where a model $f : \mathcal{X} \rightarrow \mathbb{R}$ is used to predict the response variable given an input $\mathbf{x}$, which consists of individual features $(\mathbf{x}_1, \mathbf{x}_2, \ldots, \mathbf{x}_n)$. Let $p(\mathbf{x})$ to represent the data distribution with support on $\mathcal{X} \subseteq \mathbb{R}^n$. We use bold symbols $\mathbf{x}$ to denote random variables and normal symbols $x$ to denote values. Let $\mathbf{x}_S$ and $x_S$ denote a subset of features, i.e. players in a game, and values for different $S \subseteq N$, respectively. Then, a cooperative game $\nu : \mathcal{P}(N) \rightarrow \mathbb{R}$ for estimating Shapley-based feature attributions and interactions is defined as

$$\nu(S) := f_S(x_S) := \mathbb{E}_{q(\mathbf{x}_{\bar{S}})}\big[f(x_S, \mathbf{x}_{\bar{S}})\big] = \int f(x_S, x_{\bar{S}})q(x_{\bar{S}})dx_{\bar{S}},$$

where $\bar{S} = N \setminus S$ denotes the set complement. The feature distribution $q(\mathbf{x}_{\bar{S}})$ most often considered in the literature is either a *marginal distribution* when $q(\mathbf{x}_{\bar{S}}) := p(\mathbf{x}_{\bar{S}})$ [18, 55], or a *conditional distribution* when $q(\mathbf{x}_{\bar{S}}) := p(\mathbf{x}_{\bar{S}} \mid x_S)$ [1, 26, 62].

In general, empirical estimation of a conditional feature distribution is challenging [1, 18, 55, 62]. Most recently in [63], the authors benchmark several methods for approximating SVs based on marginalizing features with a conditional distribution $p(\mathbf{x}_{\bar{S}} \mid x_S)$, without a clear best, i.e. different methods are appropriate in different practical situations. Thus, we combine the decision tree-based and sampling approaches [63] to implement a baseline *conditional imputer* in `shapiq.ConditionalImputer`. The class can be easily extended to include more algorithms, which we leave as future work. The rather standard imputation with a marginal distribution $p(\mathbf{x}_{\bar{S}})$ is implemented in `shapiq.MarginalImputer`, as well as imputation with predefined baseline values with `shapiq.BaselineImputer`. All imputers are used by the appropriate game benchmarks, and available for approximating feature interaction explanations in `shapiq.TabularExplainer` via the `imputer` parameter.

Listing 2 shows a more advanced API for setting a specific imputer and approximator in `shapiq`.[3]

```
X, model = ...
import shapiq
# create an imputer object
imputer = shapiq.ConditionalImputer(model=model, data=X, sample_size=100)
# create an approximator object
approximator = shapiq.KernelSHAPIQ(n=X.shape[1], index="SII", max_order=3)
# create an explainer object
explainer = shapiq.Explainer(model, X, imputer=imputer, approximator=approximator)
# choose a sample point to be explained
x = X[0]
# approximate feature interactions given the specificed budget
interaction_values = explainer.explain(x=x, budget=1024)
# retrieve 3-order feature interactions
interaction_values.get_n_order_values(3)
# visualize 1-order and 2-order feature interactions on a graph
interaction_values.plot_network(feature_names=...)
```

**Listing 2:** Exemplary code for defining an imputer and approximator for explanation with `shapiq`.

---

[3]For details, refer to the notebook examples at `https://shapiq.readthedocs.io` referring to imputers.

# D Guide for Interpreting Shapley Interaction Visualizations

This section provides information for interpreting visual representations of SIs, offering insights into how interactions between players—such as features in XAI or individual observations for data valuation—are depicted in network and SI graph plots. We propose two types of visualizations: the **network plot** [38, 61] and the **SI graph plot**, with the latter generalizing the former.

**General description.** In both visualizations, players are represented as nodes, while explanations in the form of interactions are depicted as edges linking these nodes. The network plot is limited to second-order interactions, meaning it only displays edges between two nodes, whereas the SI graph plot accommodates interactions of any order, with interactions involving more than two players represented as hyper-edges connecting three or more nodes. Single-order interactions are represented by the size of the nodes, with larger nodes indicating stronger main effects. The strength and direction of these interactions are encoded through the color, transparency, and thickness of the edges; stronger interactions are shown as thicker and more opaque edges, while weaker interactions are represented by thinner and more transparent edges. Consistent with established conventions from shap [55] visualizations, red indicates positive interactions, and blue indicates negative interactions. In both visualizations, nodes are drawn in a circular layout by default but can also be positioned based on a predefined graphical structure. The network plot originates in [38] to illustrate second-order interactions for global effects. It was further adapted for local SIs in [61], whereas the SI graph plot extends this concept by allowing for the visualization of higher-order interactions, thus providing a more comprehensive view of the cooperative game structure.

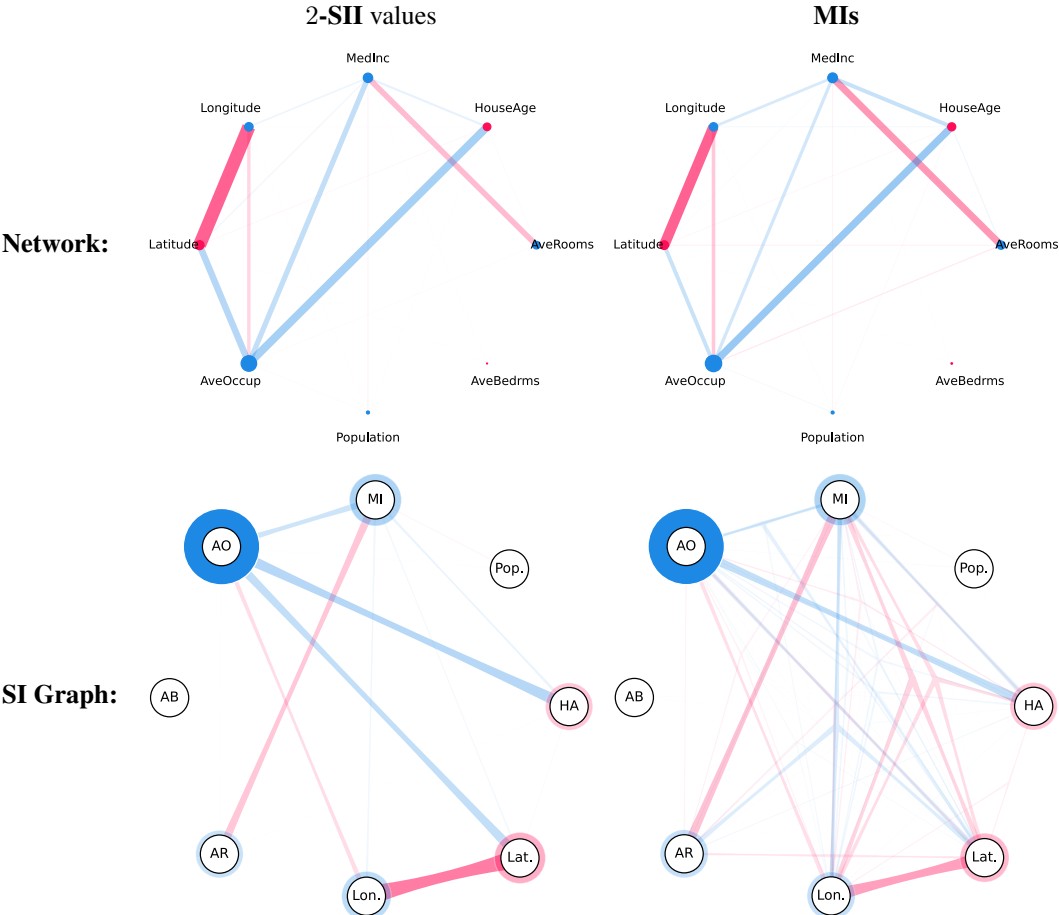

**Figure 6:** Network plots (top row) and SI graph plots (bottom row) for 2-SII scores (left) and MIs (right) as explanations for an observation from the *CaliforniaHousing* dataset and a random forest.

**Interpretation of the example visualizations.** Figure 6 shows example network and SI graph plots. We divide the *CaliforniaHousing* dataset into train and test splits. A `RandomForestRegressor` from `scikit-learn` [67] is fitted to the training data, achieving an $R^2$ score of 80%. We select an observation from the test set and compute 2-SII scores and MIs using `shapiq`'s implementation of TreeSHAP-IQ [61]. For further details regarding the dataset, training, and comparison with other visualizations, we refer to the accompanying notebook on "Visualizing Shapley Interactions" in `shapiq`'s documentation. The observation to be explained has a ground truth property value of $1.575$ (in $100,000$ USD) and is predicted to be worth $1.62$ (in $100,000$ USD). The baseline prediction of the model is around $2.071$ (in $100,000$ USD). This means that with all features provided, the model predicts the property to be worth less than the average house in the dataset. From both the network plots and SI graph plots in Figure 6, it is clear that the *AveOccup* (AO) feature has a strong negative influence on the prediction compared to the baseline, as indicated by the large blue node. However, some features and interactions have positive effects (red edges). Specifically, the interaction between *Latitude* (Lat.) and *Longitude* (Lon.), which encodes the exact location of the property, has a positive influence on the property's valuation. The MI graph plot further reveals that higher-order interactions exist, as shown by hyperedges connecting more than two features. For example, there is a sizable positive third-order interaction between *Longitude* (Lon.), *Latitude* (Lat.), and *MedianIncome* (MI). A positive fourth-order interaction involving the same three features and the *HouseAge* (HA) feature also exists.

# E   Details of the Experimental Setting and Reproducibility

This section contains additional information regarding the experimental setup and definition of the cooperative games. For further information we refer to the benchmark configuration as part of `shapiq.benchmark`.

## E.1   Generated Cooperative Games

The game-theoretical quantification of interaction demands a formal cooperative game specified by a player set $N$ and value function $\nu : \mathcal{P}(N) \to \mathbb{R}$. The players for each benchmark game are already given in Table 3, leaving the value functions open to be specified with what we catch up on here.

**Local Explanation.**   For a specified datapoint $x$, the worth of a coalition of features $S$ is given by the model's predicted value $h(x_S)$ using only the features in $S$. The features outside of $S$ are made absent in $x_S$ by imputing them with a surrogate value in order to remove their information. For tabular datasets such as *AdultCensus*, *BikeSharing*, and *CaliforniaHousing* this is done by marginal or conditional imputation. For the language model predicting the sentiment of movie review excerpts, missing words are set to the masked token. Missing pixels (patches) for the vision transformer image classifier are also set to the masked token. For the ResNet image classifier, removed superpixels are collectively set to a mean value (gray).

**Global Explanation.**   Instead of specifying a single datapoint and considering the model's output, the model's loss is averaged over a number of fixed datapoints $x_1, \ldots, x_M$. The model's loss for a coalition $S$ and datapoint $x_m$ is computed by comparing its prediction $h(x_{mS})$ with the ground truth target value. The imputation of absent features is done as for local explanations.

**Tree Explanation.**   This is a specialization of local explanations for tree models, made feasible by the capabilities of TreeSHAP-IQ to compute ground-truth SVs and SIs values, which allows the evaluation games with substantially more features. Features are imputed according to the tree distribution [54, 61]. Consequently, the worth of the empty coalition containing no features is the tree's average prediction, e.g. baseline value.

**Uncertainty Explanation.**   Similar to local explanations, the model's prediction with missing features imputed to a fixed datapoint is evaluated. Instead of referring to the predicted value, the value function is given by the prediction's uncertainty for which three measures are available: total, epistemic, and aleatoric uncertainty. Hence, the Shapley values of the features attribute their individual contribution to the decrease in uncertainty caused by their information.

**Feature Selection.**   The available data is split into a training set $\mathcal{D}_{\text{Train}}$ and test set $\mathcal{D}_{\text{Test}}$. Given a learning algorithm $\mathcal{A}$, a coalitions worth $\nu(S)$ is given by the generalization performance of the model $h_S$ on $\mathcal{D}_{\text{Test}}$ that results from applying $\mathcal{A}$ on $\mathcal{D}_{\text{Train}}$ using only features in $S$, known as *remove-and-refit*. The worth of the empty coalition is set to 0.

**Ensemble Selection.**   Replacing features in feature selection by weak learners, and adapting the learning algorithm to construct an ensemble out of those, leads to ensemble selection. Each coalition $S$ of base learners is evaluated by the performance of the resulting ensemble on a separate test set, known as *remove-and-re-evaluate*. Likewise, we set $\nu(\emptyset) = 0$.

**Data Valuation.**   Continuing in the spirit of *remove-and-refit*, a new model is fitted to each coalition of datapoints. The generalization performance of the resulting model on a separate test set is set to be the coalition's worth. The value of the empty coalition is set to 0.

**Dataset Valuation.**   The setup is analogous to data valuation, where instead of single datapoints being understood as players, the available data is partioned and each subset is viewed as a player.

**Cluster Explanation.**   Similar to feature selection, *remove-and-refit* is applied. Instead of fitting a model, a clustering algorithm forms multiple clusters on the dataset using only the available features of a coalition $S$. The worth $\nu(s)$ is given by a cluster evaluation score (see Tables 4 and 5 for details). A cluster score of 0 is assigned to the empty coalition.

**Unsupervised Feature Importance.** Given a coalition of features $S$, a set of datapoints can be understood as observations generated by a joint distribution of $S$ and used to estimate this distribution by measuring the frequencies of feature values. This, in turn, allows to measure the entropy and thus also total correlation of a subset of features which is used as the worth of $S$. Since the total correlation measures the amount of shared information, each feature's assigned Shapley value quantifies its contributed information to the group. The total correlation of the empty set is naturally 0.

**Sum of Unanimity Models (SOUMs).** A unanimity game is a synthetic game for a coalition $U \subseteq N$ with $\nu_U(T) := \mathbf{1}_{T \supseteq U}$, i.e. outputs one, if all players of $U$ are present, and zero otherwise. The sum of unanimity model (SOUM) is a linear combination of randomly sampled unanimity games. For uniformly sampled coefficients $a_1, \ldots, a_m \in [-1, 1]$ and subsets $U_1, \ldots, U_m \subseteq N$ uniformly sampled by size, where we restrict the SOUM to specific maximum subset sizes. The value function then reads as

$$\nu(T) := \sum_{\ell=1}^{m} a_\ell \nu_{U_\ell}(T).$$

For SOUMs, the MIs as well as all SIs can be efficiently computed in linear time, cf. [28, Appendix B.7].

### E.2 Computational Resources

This section contains additional information regarding the computational resources required for the empirical evaluation of this work. The main computational burden stems from pre-computing the benchmark games for $n \leq 16$ players and from running all of shapiq's SV and 2-SII approximation methods. Still, the experiments require only a modest range of computational resources. The games are pre-computed on a "11th Gen Intel(R) Core(TM) i7–11800H 2.30GHz" machine requiring around 240 CPU hours. The approximation experiments have been run on a compute cluster using 80 CPUs of four "AMD EPYC 7513 32–Core Processor" units for 24 hours resulting in about 1920 CPU hours.

### E.3 Data Availability and Reproducability

The data to the pre-computed games is available at https://github.com/mmschlk/shapiq/tree/v1. Utility functions exist in shapiq that automatically download and instantiate the games. The code for reproducing the experimental evaluation can be found at https://github.com/mmschlk/shapiq/tree/v1/benchmark and https://github.com/mmschlk/shapiq/tree/v1/complexity_accuracy.

## F    Benchmarking Analysis: Additional Results

This section contains additional experimental results. Mainly, this section contains exemplary MSE and Precision@5 approximation curves for a benchmark game of each application domain. These results can be found in Figures 9 to 11. The dataset names used for the benchmark games are abbreviated as described in Appendix B. Figure 7 shows the overview of the benchmark results additionally to Figure 5 for the mean absolute error (MAE).

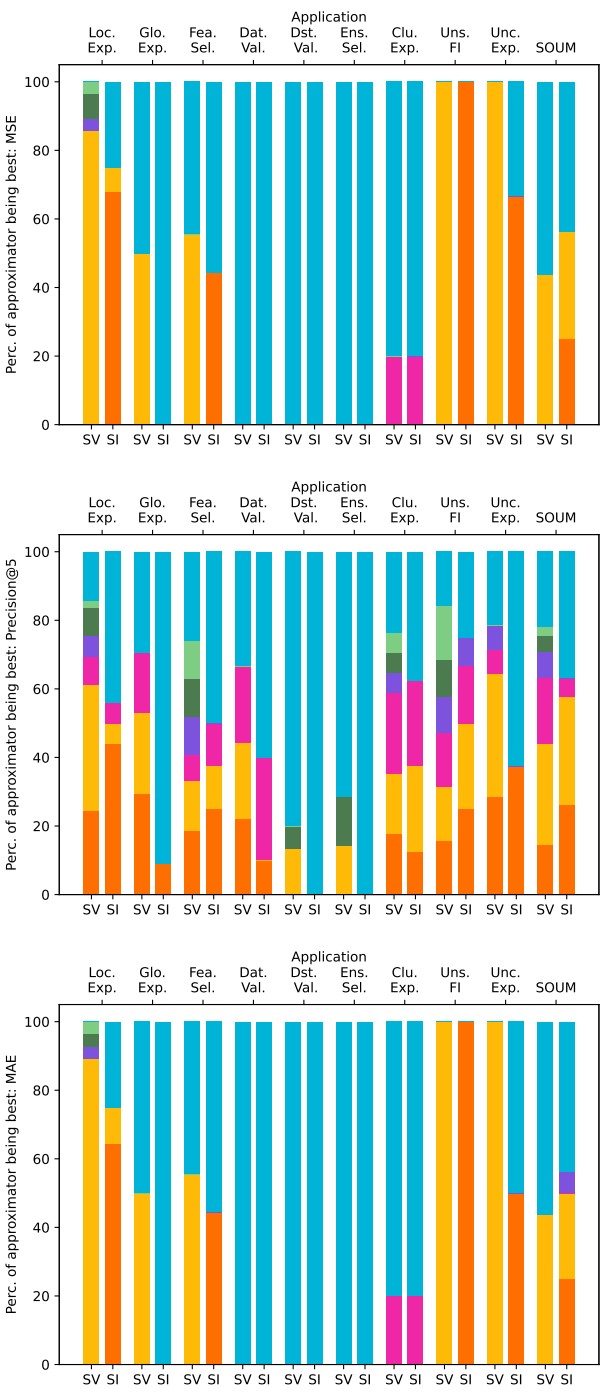

**Figure 7:** Benchmark overview approximation results for MSE (top), Precision@5 (middle), and MAE (bottom).

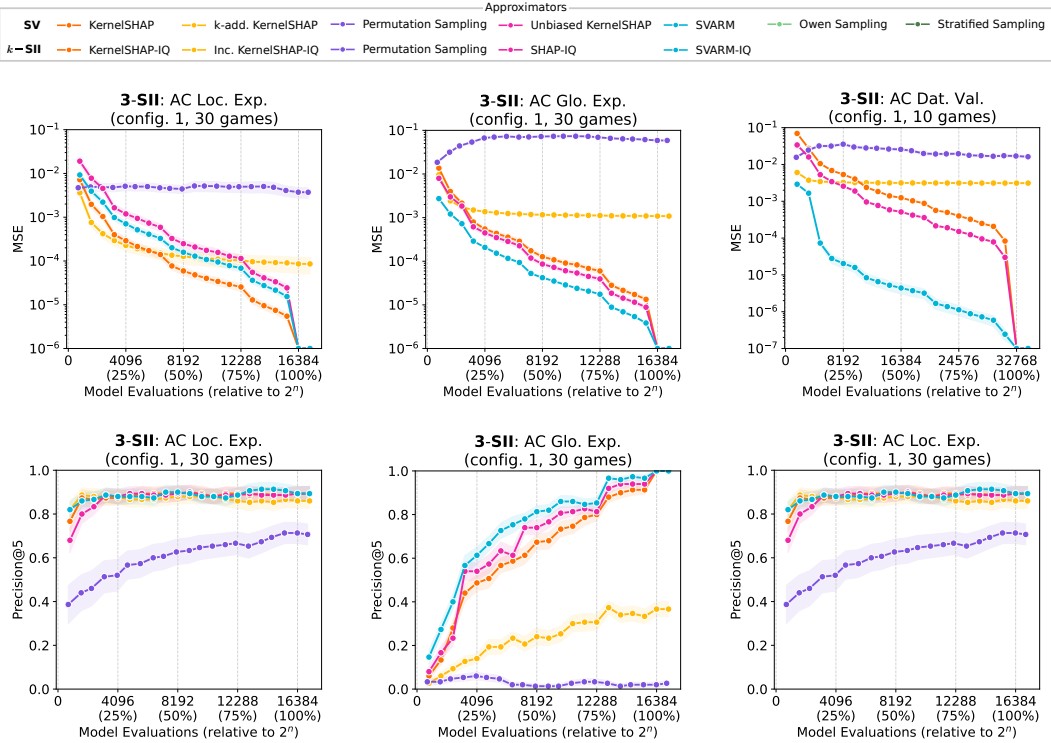

**Figure 8:** Approximation qualities in terms of MSE **(top)** and Precision@5 **(bottom)** for 3-SII higher-order interactions for three benchmark settings based on the *AdultCensus* (AC) dataset including Local Explanation **(left)**, Global Explanation **(middle)**, and Data Valuation **(right)**.

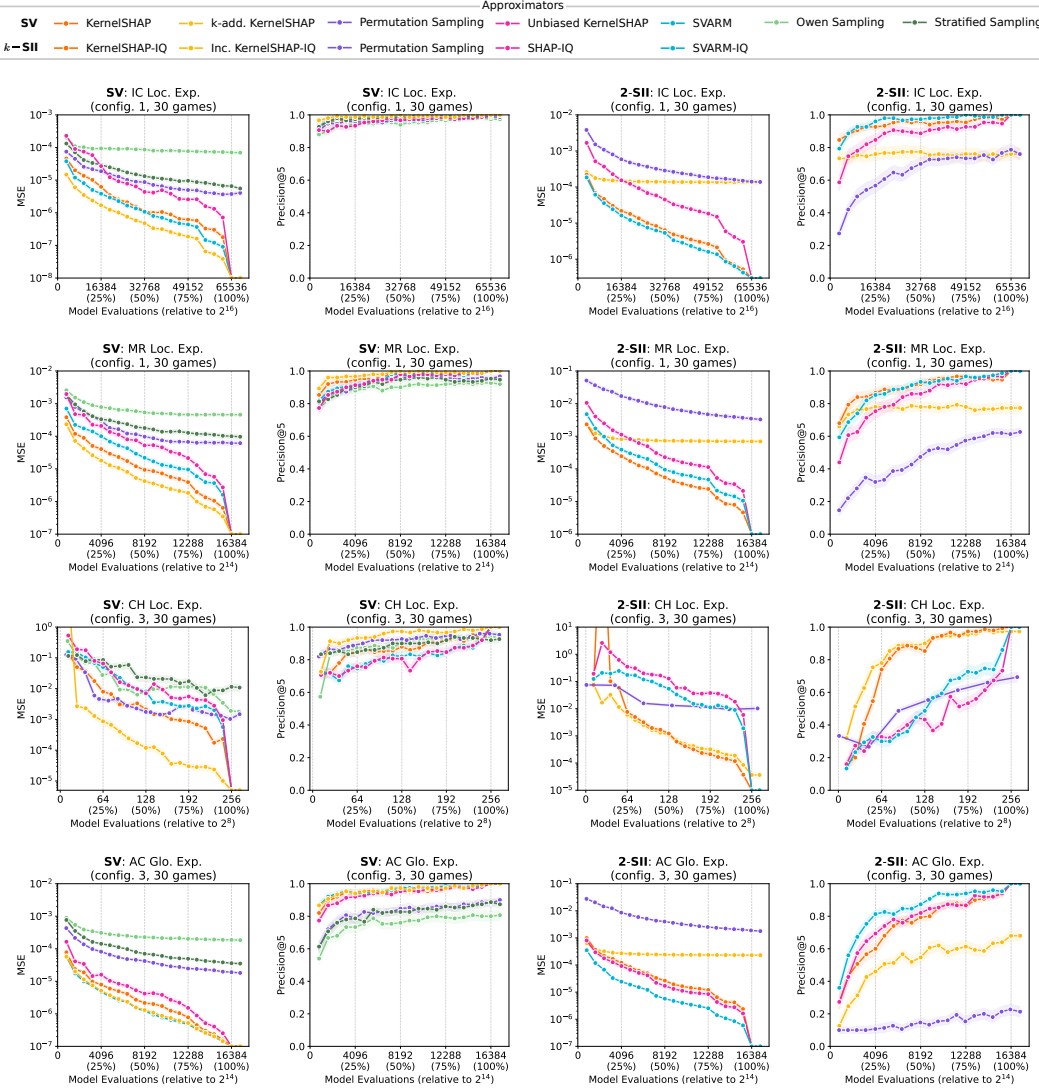

**Figure 9:** Additional SV (column one and two) and SI (column three and four) approximation results for different benchmark games from the Local Explanation (first row, vision transformer image classifier with $n = 16$ patches), Local Explanation (second row, language model predicting movie review sentiment with $n = 14$ words), Local Explanation (third row, dataset *CaliforniaHousing* with $n = 8$ features) and Global Explanation (fourth row, dataset *AdultCensus* with $n = 14$ features) domain.

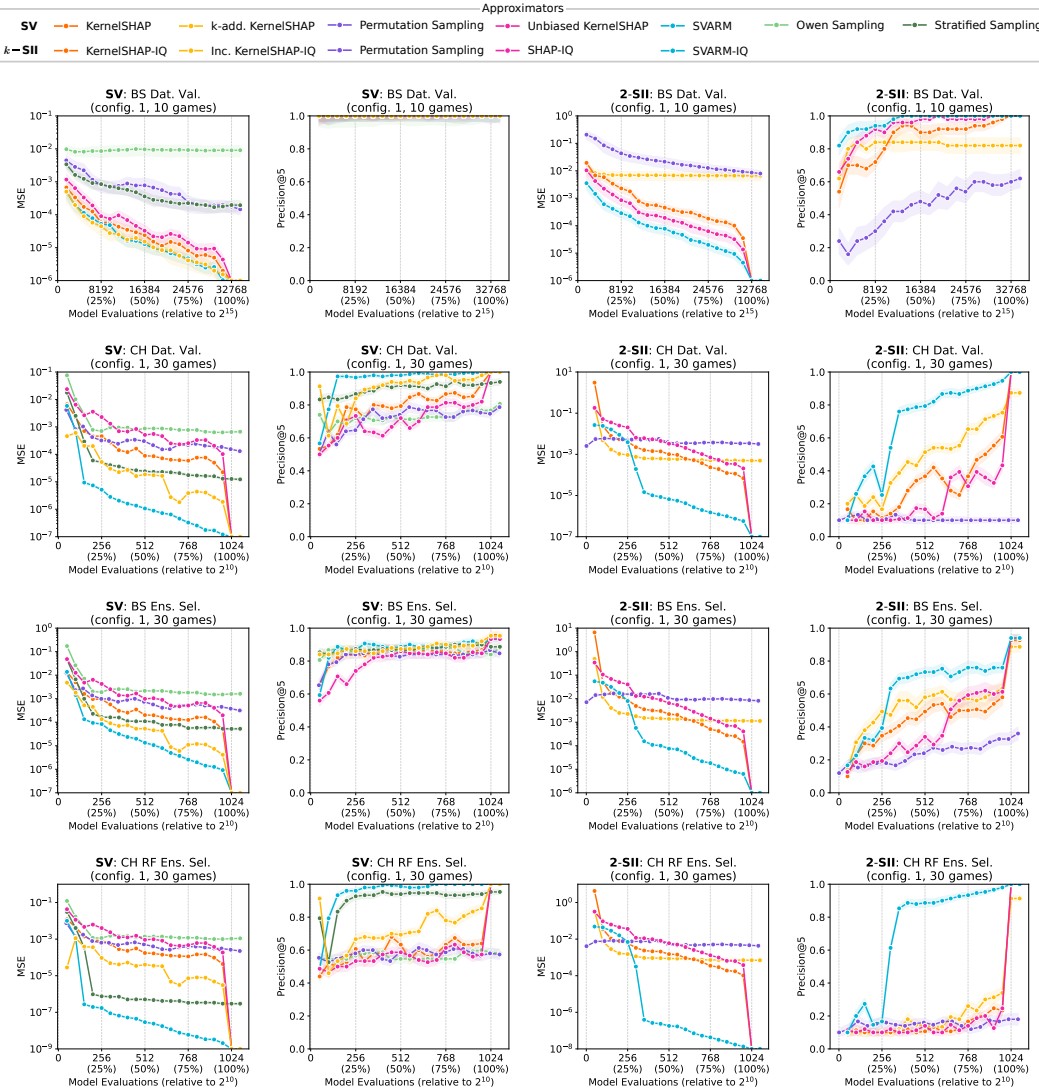

**Figure 10:** Additional SV (column one and two) and SI (column three and four) approximation results for different benchmark games from the Data Valuation (first row, *BikeSharing* with $n = 12$ features), Dataset Valuation (second row, *CaliforniaHousing* with $n = 8$ features), Ensemble Selection (third row, dataset *BikeSharing* with $n = 12$ features) and Random Forest Ensemble Selection (fourth row, dataset *CaliforniaHousing* with $n = 8$ features) domain.

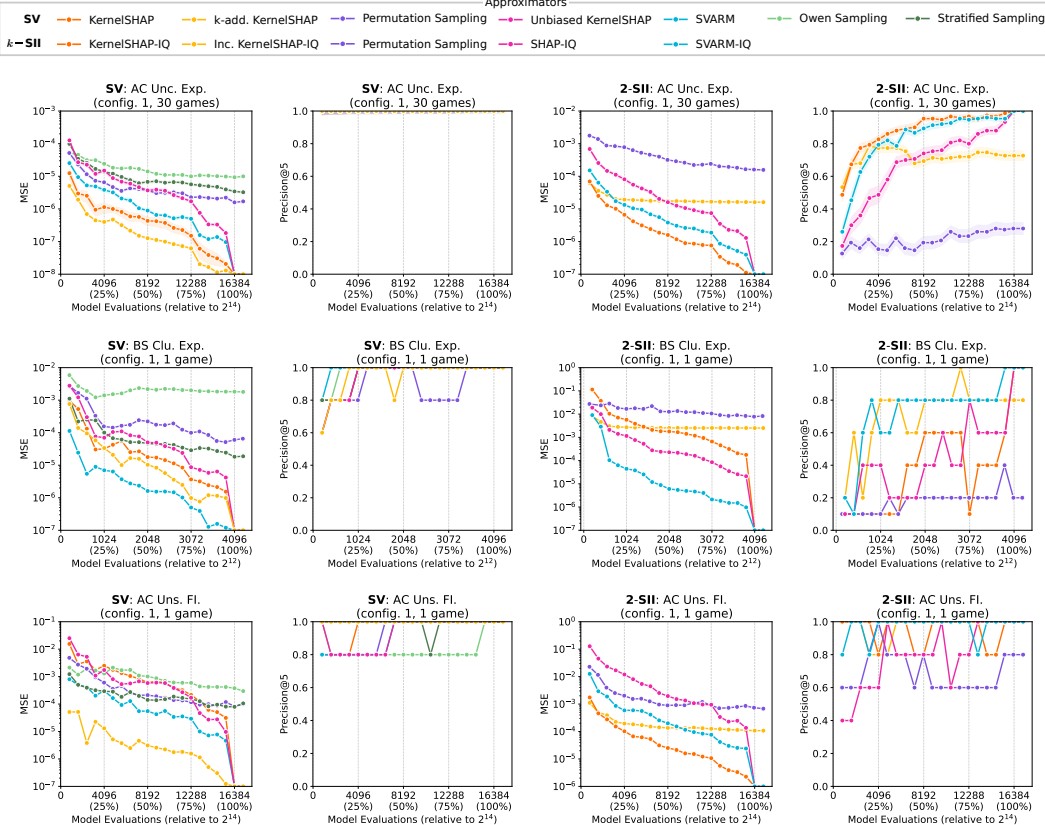

**Figure 11:** Additional SV (column one and two) and SI (column three and four) approximation results for different benchmark games from the Uncertainty Explanation (first row, *AdultCensus* with $n = 14$ features), Cluster Explanation (second row, *BikeSharing* with $n = 12$ features), and Unsupervised Feature Importance (third row, dataset *AdultCensus* with $n = 14$ features) domain.

# G  Glossary of Acronyms

