# OpenReview forum: "shapiq: Shapley Interactions for Machine Learning"
_NeurIPS.cc/2024/Datasets_and_Benchmarks_Track — NeurIPS 2024 Track Datasets and Benchmarks Poster_

### Official Review · Reviewer_78Q9 · 2024-06-25
**This paper is an open-source Python package of SHAP-IQ: Unified Approximation of any-order Shapley Interactions (NeurIPS23) with the same authors, which is designed to efficiently compute Shapley Values (SVs) and Shapley Interactions (SIs).**

**Rating:** 7
**Confidence:** 5
**Correctness:** Yep
**Clarity:** Yep

**Review:**

strong points
(s1): The introduction of shapiq as a Python package consolidates various state-of-the-art algorithms for computing Shapley Values (SVs) and any-order Shapley Interactions (SIs), making it a versatile and comprehensive tool for researchers and practitioners in machine learning.
(s2): The inclusion of a benchmarking suite with pre-computed games and ground-truth values for 11 machine-learning applications is a significant strength. This allows for the comparison of different SI algorithms across various domains
(s3): This paper is well-written and easy to follow.

weak points
(w1): Figure 4 can be annotated and interpreted in more detail.

**Strengths:**

pls see Review

**Additional Feedback:**

NA

**Documentation:**

yep

**Limitations:**

Yep

**Opportunities For Improvement:**

(d1): What does ‘size’ mean in Figure 4, and why can ‘order’ be bigger than ‘size’?
(d2): The paper mentioned data valuation, but I checked the documentation, and the tutorial all explains feature interactions of model predictions

**Relation To Prior Work:**

yep

**Summary And Contributions:**

This paper is an open-source Python package of SHAP-IQ: Unified Approximation of any-order Shapley Interactions (NeurIPS23) with the same authors, which is designed to efficiently compute Shapley Values (SVs) and Shapley Interactions (SIs). The shapiq package consolidates state-of-the-art algorithms to compute SVs and any-order SIs, offering an application-agnostic framework. It includes a benchmarking suite for evaluating computational performance across domains and supports various models, including vision transformers and gradient boosting models. The package facilitates the explanation and visualization of feature interactions, extending the use of SVs and SIs in machine learning beyond traditional feature attributions. With comprehensive documentation and examples, shapiq aims to advance research and practical applications in explainable AI.

---

> ### Author Rebuttal · Authors · 2024-08-16
>
> **We sincerly thank the anonymous reviewer** for the appreciation of our work and the feedback!
>
> You are right, with the clarity regarding Figure 4. To improve the manuscript, we will **revise the quite technical language and description regarding Figure 4** to make the intuition clearer. In general, Figure 4 describes how higher-order interactions help capture complicated relationships in cooperative games. Figure 4 (a) aims to showcase this with a simple example: _explaining_ a _single_ interaction of a particular _size_ with interactions of particular _orders_. Intuitively speaking, if we increase the order of an explanation (i.e. the Shapley interactions), we increase its _expressivity_, and the explanation can capture more nuances of the game. We measure how faithful the explanation is to the desired ground-truth interaction present in the synthetic game by reconstructing the interaction from the explanations and measuring the reconstruction error with the $R^{2}$ score. If the order of the explanations/interactions is _larger_ than the size of a ground-truth interaction in the game, there is a perfect reconstruction ($R^{2} \rightarrow 1$), and all variants of the Shapley interactions capture this. If the expressivity/order is lower than the size of an interaction in a game, there can be a non-perfect reconstruction. Reconstructing interactions of higher sizes, decreases the faithfulness of the reconstruction (it’s harder to reconstruct high-order interactions with lower orders).
>
> Thank you for the comment regarding the **data valuation tutorials**. The application of Shapley interactions to this domain offers promising opportunities for future work. Note that applications of interactions in machine learning have primarily focused on feature attribution and explanations and have not, to the best of our knowledge, been extensively investigated in domains such as data or dataset valuation. To facilitate future work, we plan to greatly improve the documentation and flexibility of `shapiq` by offering a **tutorial series** building on top of the already defined benchmark games (see also our response to reviewer D8y8). We hope that this will help practitioners and experts in specific application domains to leverage the strengths of higher-order interactions in these domains.

---

### Official Review · Reviewer_StcH · 2024-07-12
**A useful Python package for Shapley-inspired feature interactions in machine learning, with some room for improvement.**

**Rating:** 6
**Confidence:** 4
**Correctness:** The evaluation seems to make sense.
**Clarity:** The writing and presentation are larg…

**Review:**

The motivation for a benchmark including a suite of approximators for Shapley value based methods including Shapley interactions is quite meaningful. The implementation of the benchmark does seem sound. However, it does seem that the focus can be sharpened to highlight the contribution, and the suite of approximation and its considerations can be made more extensive.

Specifically, most part of the work seems to be focused on the setting of feature interaction, though there are other use cases for Shapley value (e.g., data valuation). It may be helpful to focus on feature interaction and the role of Shapley interactions, because the considerations and requirements in these settings (e.g., for feature interactions vs. data valuation would be different).

Regarding the suite of approximation methods, there are some considerations/related works that can be incorporated to strengthen this paper. For instance, the theoretical implications of the approximation of the Shapley value (Zhou et al., 2023), which has also open-sourced their implementation of several SV approximation methods.


**References**

Probably approximate Shapley fairness with applications in machine learning. Zhou, et al., AAAI 2023.

**Strengths:**

- The motivation for the benchmark for the feature interaction use case in machine learning is clear.
- The theoretical discussion on the Shapley interactions and other related concepts is appreciated.
- The implementation including various approximators and use cases seems to be useful for practitioners.

**Additional Feedback:**

In Figure 5 (b), are the colors the same as the legends for the line plots?

In the relatively smaller use cases (i.e., $n\leq 16$) the exact SI/values are computed and stored in the benchmark, which can be useful for practitioners who want design new algorithms. For the larger use cases, while the exact SI/values are difficult to compute, it may still be possible to pre-compute and store an approximation, for instance using the simplest Monte-Carlo estimation (with clearly documented sample and computational costs) which can serve a useful baseline for practitioners. In many cases, the lack of a baseline (even a possibly inaccurate one) can discourage a use from considering a benchmark.

**Documentation:**

The authors open-sourced their implementation and seems sufficient to support reproducibility.

**Ethics:**

No ethics review is required.

**Limitations:**

The authors describe the limitations in Section 5, which appear to make sense.

**Opportunities For Improvement:**

As mentioned in the review above,

1. The clarity and focus of the paper can be improved. One possibility is to focus specifically on feature interactions in machine learning and the implications of Shapley and other game theoretic solution concepts in this use case.

    In lines 21-25,
    > Assigning value to entities collectively performing a task is essential in various real-world applications of machine learning (ML) [59, 70]. For instance, when reimbursing data providers based on the value of data [29, 75], or justifying a model’s prediction based on value of feature information [12, 17, 18, 24 51, 72]. The fair distribution of value among a group of entities is a central aspect of cooperative game theory ...

    While I agree that these different use cases do adopt the Shapley value or some variant, the rest of the paper and exposition seems to be largely focused on the feature interaction use case, which can make the exposition on other use cases seem less extensive. Furthermore, these use cases do have differing requirements on the solution concepts, which may thus require different treatment when it comes to approximation and implementation. For instance, it has been argued that the so-called _efficiency_ property (i.e., the sum of all Shapley values equals a predetermined constant such as $1$) is _not_ necessary or desirable for the data valuation use case (Kwon and Zou, 2022 and Zhou et al., 2023). The implication is a more general and less restrictive definition of the Shapley value (not required to satisfy the efficiency), which is identical up to a positive linear scaling factor.


2. The implementation of various approximators certains seems appealing to a practitioner. One opportunity for improvement is the implication of these approximators, specifically with respect to the properties of the Shapley value (or its variant). This is in addition to Section 4.1.

    In lines 29-30
    > The SV fairly distributes the overall worth among individuals by evaluating the game for all coalitions. However, it does not give insights on synergies or redundancies between entities.

    Previously, Zhou et al., (2023) examined how the approximation affects the _fairness_ of the Shapley value approximates in their proposed probabily approximate fairness framework, which does seem to be relevant when comparing different approximators. Moreover, this paper mentions that the vanilla Shapley value may be lacking when it comes to providing insights on synergies or redundancies, which is also a relevent point. Hence, it may be helpful to formally describe how the approximation affects the ability of Shapley interactions to provide such insights, for instance so that a practitioner can compare and select the approximators for their use cases.


**References**

Beta Shapley: a Unified and Noise-reduced Data Valuation Framework for Machine Learning. Kwon and Zou. AISTATS 2022.

Probably approximate Shapley fairness with applications in machine learning. Zhou, et al., AAAI 2023.

**Relation To Prior Work:**

The relation to prior work is discussed in Section 1, and mostly sufficient. Some useful references (which seem to be missing) are mentioned in the review/opportunities for improvement parts.

**Summary And Contributions:**

The authors provide a Python library that implements a general approximation interface for Shapley interaction (SI) algorithms and the explanation and visualization of SI in machine learning, contains a benchmark of various use cases for the SI and include empirical evaluation of approximators.

---

> ### Author Rebuttal · Authors · 2024-08-16
>
> We **gratefully thank** the anonymous reviewer for their appreciation of our contribution and their engagement with our work!
>
> We welcome your idea for larger use-cases with $n > 16$ or even $n \gg 16$. We could easily store some estimates of the approximations as _pseudo ground truth values_. A selection of readily available approximation results could definitely be helpful! Future work could even investigate if estimates by _different_ approximators can be combined into an aggregated estimate with a potentially higher quality.
>
> Thank you for spotting the missing description in Figure 5. Yes, the colors of (b) are the same as the legend of the line plots. We will fix this.
>
> >Regarding the suite of approximation methods, there are some considerations/related works that can be incorporated to strengthen this paper. For instance, the theoretical implications of the approximation of the Shapley value (Zhou et al., 2023), which has also open-sourced their implementation of several SV approximation methods.
>
> Thank you for pointing out the related work of Zhou et al. (2023), which we were unaware of before. We will definitely add this reference in a revised version of our work.
>
> >The clarity and focus of the paper can be improved. One possibility is to focus specifically on feature interactions in machine learning and the implications of Shapley and other game theoretic solution concepts in this use case. In lines 21-25, [...] While I agree that these different use cases do adopt the Shapley value or some variant, the rest of the paper and exposition seems to be largely focused on the feature interaction use case, which can make the exposition on other use cases seem less extensive.
>
> Our main focus in the paper is the `shapiq` software with its general applicability to well-established benchmark approximation algorithms and capability to explain machine learning models. We certainly observe that Shapley-based feature interactions are currently the most exposed researched direction due to their impact on explainability. However, we hope that the implemented algorithms and benchmarks in `shapiq` will further facilitate research on Shapley interactions for other use cases, including data valuation, feature selection, or ensemble valuation. As also mentioned in our response to reviewer D8y8, we aim to offer a tutorial series as part of the documentation that covers the application of `shapiq` in different application scenarios. We hope that this will facilitate researchers unfamiliar with Shapley interactions to spot the benefit in their respective subject areas.
>
> >Furthermore, these use cases do have differing requirements on the solution concepts, which may thus require different treatment when it comes to approximation and implementation. For instance, it has been argued that the so-called efficiency property (i.e., the sum of all Shapley values equals a predetermined constant such as 1) is not necessary or desirable for the data valuation use case (Kwon and Zou, 2022 and Zhou et al., 2023). The implication is a more general and less restrictive definition of the Shapley value (not required to satisfy the efficiency), which is identical up to a positive linear scaling factor.
>
> Yes, you are absolutely right! We agree that different use cases might require different approximator methods. Specifically, loosening the efficiency property may be an important prerequisite for certain application domains such as data valuation. Other semivalues like the Banzhaf value are definitely an option. This is exactly the reason why we built `shapiq` in its general manner and include a wide range of game theoretic concepts. Next to the `ExactComputer`, estimators that build on the notion of _Cardinal Interaction Indices_ like SHAP-IQ [27] and SVARM-IQ [43] can be used to estimate a wide range of indices like the Banzhaf value or the Banzhaf Interaction index. Faithfull versions of this are also possible via the FSII Regressions [76].
>
> >The implementation of various approximators certains seems appealing to a practitioner. One opportunity for improvement is the implication of these approximators, specifically with respect to the properties of the Shapley value (or its variant). This is in addition to Section 4.1. In lines 29-30 [...] Previously, Zhou et al., (2023) examined how the approximation affects the fairness of the Shapley value approximates in their proposed probabily approximate fairness framework, which does seem to be relevant when comparing different approximators. Moreover, this paper mentions that the vanilla Shapley value may be lacking when it comes to providing insights on synergies or redundancies, which is also a relevent point. Hence, it may be helpful to formally describe how the approximation affects the ability of Shapley interactions to provide such insights, for instance so that a practitioner can compare and select the approximators for their use cases.
>
> This certainly is related, as Shapley interactions reveal the synergies and redundancies that can get obfuscated in the aggregation of the Shapley value. We hope that by providing `shapiq` and a tailored documentation for practitioners or experts in particular research domains, we will lower the barrier of entry and facilitate novel research in adjacent domains such as fairness.

---

### Official Review · Reviewer_Z6Aj · 2024-07-22
**Presents a new open source package for benchmarking and applying a wide range of implementations of "Shapley values" and "Shapley interactions" with implications for feature attribution and observation attribution**

**Rating:** 9
**Confidence:** 4
**Correctness:** Implementation of benchmarking seems …
**Clarity:** Clarity is good through.

**Review:**

Overall, this is a strong contribution to the D&B track. The paper itself seeks to bring additional "unity" to several active research areas through the introduction of a shared library / schema / set of benchmarks. This type of submission has the potential to have large impact within those sub-communities, and beyond (more below on this). As described further below, quality is very high, and while there may be room for minor improvements in clarity both the paper and code are relatively clear. I think many readers will likely find this submission to make a large contribution.

**Strengths:**

### Significance
This work is likely to make a large contribution to research in explainable AI, model selection, data valuation, etc. Efforts like this to unify benchmarking and implementation can support future work along these lines. While an extension of past work in a similar vein (the `shap` package), the paper describes clear additional contributions here.

### Relevance to broad research community
In the near term, the work will likely be most impactful on researchers actively working in relevant spaces. One ideal user of the system is a researcher who wants to apply one the feature/data value definitions in a new context and would benefit from a robust set of existing benchmarks. Additionally, researchers who are working on defining modified value definitions (e.g., some new Shapley value variant) and want a standardized reference library for comparison will benefit.

In the long run, this work may be impactful based on the overall adoption of the included techniques. For instance, if Shapley values or interactions are widely used for data-related remuneration, this work would have wide impact.

### Quality
Focusing primarily on the benchmarking part of the work, the decisions here seem reasonable. The Appendix answers many possible readers questions about, "What about this additional comparison between methods?" Some important details are left to the Appendix (e.g. dataset details) but of course some tradeoffs need to be made.

The code itself (a key part of the contribution) appears well organized (with the caveat that of course there's some subjectivity here, and the goal is not to provide a user study of this new library).
### Ethical and social implications
While there are some ethical discussions to be had about some of the topics here (in particular: impacts of using explainability based methods in practice, implications of using data valuation methods to allocate resources or credit) I think for this kind of focused package/benchmark contribution, the level of engagement is reasonable. As I'll note below, space is already at a premium here given the density of content.

**Additional Feedback:**

Thanks to the authors for their contribution.

**Documentation:**

Not a dataset. In terms of code, documentation is a strength.

**Ethics:**

As noted above -- with more room, the paper could briefly touch on the implications of attribution methods being used in practice, but the current draft is reasonable to not focus on this.

No major issues with any of the D&B ethics flags.

**Limitations:**

The current is reasonable about its scope overall. One minor suggestions might be to further justify the datasets chosen, and any limitations they might bring to the table (especially e.g. the use of Census https://proceedings.neurips.cc/paper_files/paper/2021/hash/32e54441e6382a7fbacbbbaf3c450059-Abstract.html)

**Opportunities For Improvement:**

Above, I noted impact, broad relevance, and quality were all major strengths. The major opportunity for improvement with this paper is that I think the manuscript itself could be even more focused on the specific contribution of library design / unification.

There's a lot of material to cover here: the paper provides some history on similar benchmarking efforts, then covers the theoretical background of, and motivation for, using cooperative game theory and value functions in the context of ML. A mathematical description of some of the more popular approaches to assign values is provided. However, I expect that readers will end up wanting to (re)visit the relevant introductory work anyway, especially to understand the Shapley interactions and interaction indices more generally in the context of ML.

To summarize, I think it may be beneficial to reorganize the paper in a way that actually shortens Section 2 (or treats itself more as an intuitive explainer on values and interaction indices) and moves more of the theoretical background to the already existing appendix section. Overall, this kind of thing is just a subjective organizational choice and I don't feel too strongly about it. The best way to go here may depend on the intended reader (familiar readers may appreciate seeing something like Table 2 before seeing an example script; less familiar readers may prefer things reversed).

The inclusion of theoretical background is of course helpful and will help interested researchers to have greater confidence in the correctness of the work and serious technical engagement required to implement all these methods. But the cost is some researchers who are new to the space may miss the contribution of "you can quickly get started trying out all these methods in practice".

**Relation To Prior Work:**

Good discussion of work that is similar in both topic and spirit. Section 2 is useful in general for future authors of this kind of "library/benchmark" contribution.

**Summary And Contributions:**

This paper presents a new Python coding package that aims to comprehensively capture a number of approaches for computing Shapley values and "Shapley Interactions" in an application-agnostic way, such that a number of different approaches to calculating attribution scores can be used for both feature attribution, observation-level attribution, dataset attribution, etc.

---

> ### Author Rebuttal · Authors · 2024-08-16
>
> **We sincerely thank the anonymous reviewer** for the **great appreciation of our work and its potential impact**! We value your **detailed feedback**!
>
> Regarding your comments on the **theoretical background section**, you are right! An extensive description of the theoretical concepts might lead to some readers missing that you can easily use the software to work on Shapley interactions for machine learning. Yet, as you also point out, the main motivation for incorporating a short primer on the game theoretic concepts, especially the discussion of the Generalized Values and Shapley Interactions, is to make practitioners or experts in certain application domains familiar with the vast array of available concepts. This is, in our view, a valuable contribution, as depending on the domain, only the Shapley or Banzhaf values are common. One option of course coulde be to move more of this background into the appendix or loosening the technical language and amount of formal notation. However, given an additional page, we would like to **extend Figure 1** and include another horizontal element showcasing a **short example listing** similar to Figures 2 and 3. We would embed a short description of this new code snippet in the _Contribution_ paragraph. By extending the rest of the manuscript with more information on how to interact with the library the theoretical background section become (at least relatively) shorter. We hope these changes can somewhat alleviate the problem you have described without removing this small overview.
>
> Thank you for pointing out the potential **limitations of the datasets** and, in particular, the ethical concerns with the Adult Census benchmark data! Since the adult census dataset has been used in many related works, we also included it in this benchmark and software suite. However, we see that offering this dataset in our packaged version might do more harm than good and opted to remove it in future releases, where we anyways plan to incorporate a more diverse set of additional datasets.

---

### Official Review · Reviewer_D8y8 · 2024-07-25
**Identify any order of shapley interaction**

**Rating:** 7
**Confidence:** 4
**Correctness:** The package is constructed in a sound…

**Review:**

Quality

The quality of the research is good, providing a well-constructed package and comprehensive benchmarking. The authors have demonstrated thorough evaluation of various SI approximation methods across multiple datasets and tasks. The package is robust and versatile, supporting a wide range of applications in ML. The experimental design is sound, with clear metrics for assessing computational performance.

Clarity

The paper is well-written, presenting its motivation, methodology, and findings clearly. The structure is logical, and the flow of information is easy to follow. Some sections, such as the dataset descriptions and specific configurations of the models used, could benefit from additional detail for enhanced clarity.

Originality

The work is highly original, addressing a significant gap in current ML research by providing a unified framework for computing and benchmarking SIs. The inclusion of diverse datasets, tasks, and SI algorithms represents a substantial advancement in the field. The package's novel insights into higher-order feature interactions and their computational efficiency are particularly noteworthy.

Significance

The significance of the work is great, as it provides a valuable resource for the ML community. The package enables more realistic and rigorous evaluation of SI algorithms, which is crucial for advancing ML explainability.

**Strengths:**

Comprehensive Package: "shapiq" is a well-structured and versatile package that supports efficient computation and benchmarking of SVs and SIs across diverse ML applications.

Robust Evaluation: The authors perform thorough evaluation of multiple SI approximation methods, providing valuable insights into their performance across different tasks.

Novel Insights: The package offers novel insights into the computational efficiency of higher-order feature interactions and their implications for ML explainability.

**Additional Feedback:**

N/A

**Clarity:**

The paper is well-written, with a logical structure and clear presentation of ideas.

**Documentation:**

The package documentation is thorough, covering data collection, organization, and the format used. The paper includes a URL for accessing the package and mentions the availability of the codebase.

**Limitations:**

The authors have addressed some limitations of their work, such as the need for improved visualization of higher-order feature interactions and the potential for misinterpreting SIs based on the chosen index.

**Opportunities For Improvement:**

Scalability: Explore the scalability of the package on various problems.

**Relation To Prior Work:**

The paper provides a clear discussion of how this work differs from previous contributions.

**Summary And Contributions:**

The paper presents "shapiq," an open-source Python package for computing Shapley Values (SVs) and Shapley Interactions (SIs) in machine learning (ML). The package unifies state-of-the-art algorithms to efficiently compute SVs and any-order SIs, offering an application-agnostic framework. The contributions include a benchmarking suite for evaluating SIs across 11 ML applications, pre-computed games and ground-truth values for systematic assessment, and visualization tools for explaining feature interactions in model predictions. The package aims to extend the use of SVs and SIs beyond feature attributions, facilitating future research in ML explainability.

---

> ### Author Rebuttal · Authors · 2024-08-16
>
> We would like to **sincerely thank the reviewer** for acknowledging the quality and significance of our work! We definitely agree that exploring the scalability of the computation beyond our benchmark is a promising future work direction. To further facilitate this, we are continuously extending the documentation. Specifically, we aim to provide a **tutorial series** catered towards users that are unfamiliar with the game-theoretic background but knowledgeable about individual application domains in the benchmark. We hope that this can be used as a reference point for newcomers and showcase how simple it is to incorporate Shapley interactions in different subject areas. This tutorial series will also include a guide on “How to write your own custom games/value functions.” To improve the runtime of the different algorithms, **parallelization** can also be applied. For this, we already have a working example notebook in the documentation, which is, to be fair, still quite rudimentary for now. Given the level of abstractions with `explainers`, `approximators`, and `games`, parallelization can be employed on different levels.

---

> > ### Comment · Reviewer_D8y8 · 2024-08-16
> >
> > I believe incorporating tutorials and enhancing parallelization in the future plans would greatly benefit your work.
> >
> > Since this is beyond the scope of the current paper, I suggest we stick to accepting it with the current score.

---

### Author Rebuttal · Authors · 2024-08-16

**We gratefully thank the anonymous reviewers** for their time and effort in reviewing our manuscript! We are very happy about the appreciation of our work and the valuable feedback!

---

### Decision · Program_Chairs · 2024-09-26

**Decision:**

Accept (Poster)

**Comment:**

The paper presents shapiq, an open-source Python library for computing Shapley value and Shapley Interactions across several machine learning domains. It is well-written and easy to follow, with clear novel contributions to the community. It fits the benchmarking track theme perfectly. I think it is a clear accept.